# EMERGENCE OF META-STABLE CLUSTERING IN MEAN-FIELD TRANSFORMER MODELS

**Giuseppe Bruno**[1*]  **Federico Pasqualotto**[2]  **Andrea Agazzi**[1*]
[1]Department of Mathematics and Statistics, University of Bern
[2]Department of Mathematics, University of California, San Diego

## ABSTRACT

We model the evolution of tokens within a deep stack of Transformer layers as a continuous-time flow on the unit sphere, governed by a mean-field interacting particle system, building on the framework introduced in (Geshkovski et al., 2023). Studying the corresponding mean-field Partial Differential Equation (PDE), which can be interpreted as a Wasserstein gradient flow, in this paper we provide a mathematical investigation of the long-term behavior of this system, with a particular focus on the emergence and persistence of meta-stable phases and clustering phenomena, key elements in applications like next-token prediction. More specifically, we perform a perturbative analysis of the mean-field PDE around the iid uniform initialization and prove that, in the limit of large number of tokens, the model remains close to a meta-stable manifold of solutions with a given structure (e.g., periodicity). Further, the structure characterizing the meta-stable manifold is explicitly identified, as a function of the inverse temperature parameter of the model, by the index maximizing a certain rescaling of Gegenbauer polynomials.

## 1 INTRODUCTION

The transformers architecture (Vaswani, 2017), widely used in large language models, has played a pivotal role in fueling the recent AI revolution. A key element of this architecture are self-attention modules (Bahdanau, 2014), allowing to capture long-range relationships within the data, e.g., in sentences where the number of tokens is large. However, despite the impact of transformers and the key role played by this mechanism, the theoretical understanding of these models is still in an embryonic stage. In particular, a mathematical description of the representations developed by these models, and of the effect of different choices of hyperparameters on such representations is still lacking.

Recently, the work (Geshkovski et al., 2023) introduced a simplified model for the dynamics of tokens through the layers of the transformer architecture. This model consists of a nonlinear Partial Differential Equation (PDE) modeling the evolution of the tokens – interpreted as $N$ interacting particles – through the layers of the network. As showcased by the authors in their work, this model displays clustering dynamics of the tokens through the layers, a feature indicating the emergence of representations in the network (e.g., the emergence of concepts/consensus in next-word predictions).

Despite its simplicity, this model – which, in its simplest form, results from the choice of specific value, key and query matrices in the self-attention mechanism – displays a number of dynamical properties that have not been fully understood. In particular, while the mathematical description of the emergence of clusters was proven by studying the long-time (infinite depth) behavior of the token dynamics, which can be shown to generically converge to a single cluster as time (depth) goes to infinity, numerical simulations show that this phase of total collapse is preceded by several phases of partial clustering, which long-time results fail to characterize. Characterizing this phenomenon is crucial for understanding how representations develop in deep models: on the one hand, deep models are expected to produce more diverse and rich representations of the data, leading to improved generalization and performance. On the other hand, due to the finite depth of transformer models and the prohibitive computational and memory cost required to reach the single-cluster regime,

---

[*]Part of this work was carried out while at the University of Pisa.
Contacts: {giuseppe.bruno,andrea.agazzi}@unibe.ch,  fpasqualotto@ucsd.edu

these intermediate clustering phases are likely to play a crucial role in the model's effectiveness, highlighting the need for a deeper understanding of their meta-stable behavior.

**Contributions**  In this paper we study the clustering dynamics of the model developed in (Geshkovski et al., 2023) in the intermediate time range, i.e., between initialization and the emergence of the first set of clusters. To do so, we give a detailed mathematical description of the dynamics of the network in the large-context regime, i.e., when the number of tokens on which the transformer acts is large, characterizing the intermediate, meta-stable phase of the network before it collapses to the single-cluster limit. More specifically we prove the following:

1. We rigorously prove the convergence, as the number of tokens $N$ becomes large, to the dynamics of the limiting law under the mean-field PDE. We further show that this convergence is non-uniform in time, justifying the more refined metastability analysis carried out below.

2. We characterize the evolution of the $N$-particle measure close to initialization, identifying an initial phase where a certain number of clusters starts to form. We explicitly characterize the structure of the solution forming in this first phase – e.g., its periodicity – as a function of the temperature parameter, the number of tokens and the embedding dimension $d$.

3. We perform an in-depth analysis of the dynamics on longer time scales to prove that the periodicity developed in the first phase is maintained over time intervals of length $O(\ln N)$. This proves the existence of a meta-stable phase where the network learns a richer representation of the data, before the total collapse phase predicted by (Geshkovski et al., 2023).

We note that the above characterization has important practical implications, as it clarifies the relationship between hyperparameters such as the temperature parameter, the number of tokens or the embedding dimension and the emergence of representations in transformers by providing quantitative estimates on the "richness" of such representation and on the depth required to achieve it.

**Related works**  This paper is closely related to the works (Sander et al., 2022; Geshkovski et al., 2023), where the model we study here was developed. In (Geshkovski et al., 2023), the authors highlight the problem of investigating the dynamic meta-stability of such models, a problem that, to the best of our knowledge, remains open to this day. In this setting, the clustering phenomenon was studied in (Markdahl et al., 2017) in $d > 2$ and in (Criscitiello et al., 2024) for $d = 2$. These works give sufficient conditions for the occurrence of the collapse of the dynamics for finite number $N$ of tokens to a single cluster, which occurs with probability 1 over the initial conditions of the system. These results, however, only hold in the $t \to \infty$ limit, while our work provides a more detailed analysis, in the large $N$ limit, capturing the meta-stable properties of the dynamics as it approaches slow manifolds of intermediate representations in the form of structured solutions before reaching the ultimate single-mode collapse. The dynamic meta-stability of mean-field transformers was also studied in the complementary setting of well-separated configurations in (Geshkovski et al., 2024a).

The approach taken in this paper is closely connected to the one developed in (Chen et al., 2018; E, 2017), where the dynamics of the network's state through its depth are modeled by a differential equation. In our case, however, the state of the network can be broken down in $N$ "particles" (as opposed to one in (Chen et al., 2018)) that interact in a mean-field way through a nonlinear PDE. More generally, this paper connects to a vast literature on mean-field models for neural networks, pioneered by the papers (Rotskoff & Vanden-Eijnden, 2022; Mei et al., 2018; Chizat & Bach, 2018), later extended to more general settings and more quantitative estimates (e.g., in (Agazzi & Lu, 2021; De Bortoli et al., 2020)). While those papers also model the state of the network as an empirical measure whose evolution is described by a mean-field PDE, in this series of works the dynamics describe the training of the model and not the evolution of the model's state through its depth. Furthermore, in these cases the propagation of chaos estimates are established on bounded time intervals, while we extend these estimates on $\mathcal{O}(\log N)$ time intervals.

Another line of related works studies the fluctuations around the mean-field limit of nonlinear PDEs such as the one discussed in this work. In a more general setting, (Carrillo et al., 2020) analyzed the connection between Fourier coefficients of a given potential and the stability of the homogeneous steady state for general McKean-Vlasov equations on the torus, (Lancellotti, 2009) characterized the fluctuations in the linear regime, and (Grenier, 2000) devised a general method for investigating the instability of the Euler and Prandtl equations.

**Structure of the paper** In Section 2 we introduce the framework and some notation. In Section 3 we present the large-$N$ convergence results. In Section 4 we discuss the long-time analysis of the dynamics and the corresponding meta-stability results. In Section 5 we present some numerical simulations and in Section 6 the conclusions.

## 2 FRAMEWORK AND NOTATION

We start by introducing the models under consideration in this paper. These were first derived in (Sander et al., 2022; Geshkovski et al., 2023; 2024b), by considering the transformer architecture as a discrete-time dynamical system describing the evolution of $N$ *tokens* $\{x_i(t)\}_{i=1,\dots,N}$, given by

$$
\begin{cases} x_i(k+1) = \mathcal{N}\left(x_i + \frac{1}{Z_{\beta,i}} \sum_{j=1}^{N} e^{\beta\langle Qx_i(k), Kx_j(k)\rangle} V x_j(k)\right), & k = 0, \dots, L-1 \\ x_i(0) = x_i \end{cases}, \quad (1)
$$

where $\mathcal{N} : \mathbb{R}^d \to \mathbb{S}^{d-1}$ denotes the normalization operator and $Z_{\beta,i}$ is a normalization factor. The dynamics depend on (matrix) parameters $Q, K, V$ (query, key, value) whose significance is inherited from the transformer architecture. For simplicity, (Geshkovski et al., 2023) set $Q = K = V = Id$ and introduce the following continuous-time, simplified model problem describing the dynamics of $x_i(t) : [0, \infty) \to \mathbb{S}^{d-1}$ as a (layer-wise) limit of (1) in the spirit of (Chen et al., 2018):

$$
\dot{x}_i(t) = P_{x_i(t)}\left(\frac{1}{Z_{\beta,i}(t)} \sum_{j=1}^{N} e^{\beta\langle x_i(t), x_j(t)\rangle} x_j(t)\right). \quad \text{(SA)}
$$

Here, $P_x y$ is the projection onto $T_x \mathbb{S}^{d-1}$ given by $P_x y = y - \langle x, y\rangle x$, $\langle \cdot, \cdot \rangle$ denotes the Euclidean inner product in $\mathbb{R}^d$, $Z_{\beta,i}(t)$ is a layer-wise normalization term defined by $Z_{\beta,i}(t) = \sum_{k=1}^{n} e^{\beta\langle x_i(t), x_k(t)\rangle}$ and $\beta > 0$ is a positive constant, identified throughout as the *inverse temperature*. They also introduce a related model problem[1]

$$
\dot{x}_i(t) = P_{x_i(t)}\left(\frac{1}{N} \sum_{j=1}^{N} e^{\beta\langle x_i(t), x_j(t)\rangle} x_j(t)\right), \quad \text{(USA)}
$$

modifying the normalization factor of the sum in (SA).

It is convenient to parametrize the dynamics (SA) and (USA) by "modding out" permutation invariance. This is classical and can be achieved by setting $\mu(t) := \frac{1}{N} \sum_{i=1}^{N} \delta_{x_i(t)}$, where we denote by $\delta_x$ the Dirac delta distribution at the point $x$. This yields the continuity equation[2]:

$$
\begin{cases} \partial_t \mu + \operatorname{div}(\chi[\mu]\mu) = 0 & \text{on } \mathbb{R}_{\geq 0} \times \mathbb{S}^{d-1}, \\ \mu_{|t=0} = \mu(0) & \text{on } \mathbb{S}^{d-1}. \end{cases} \quad (2)
$$

In the above formula, $\chi[\mu] : \mathbb{S}^{d-1} \to T\mathbb{S}^{d-1}$ is given by either $\chi_{\text{SA}}$ or $\chi_{\text{USA}}$, respectively defined as

$$
\chi_{\text{SA}}[\mu](x) = P_x\left(\frac{1}{Z_{\beta,\mu}(x)} \int e^{\beta\langle x,y\rangle} y \, \mathrm{d}\mu(y)\right) \quad \text{(SA-MF)}
$$

$$
\chi_{\text{USA}}[\mu](x) = P_x\left(\int_{\mathbb{S}^{d-1}} e^{\beta\langle x,y\rangle} y \, d\mu(y)\right). \quad \text{(USA-MF)}
$$

where we used $Z_{\beta,\mu}(x) = \int e^{\beta\langle x,y\rangle} d\mu(y)$.

Note that the dynamics are then interpreted as a flow map between probability measures on $\mathbb{S}^{d-1}$. In addition, as shown in (Geshkovski et al., 2023), (2) admits a Wasserstein gradient flow structure.

Noting that (2) evolves both empirical and absolutely continuous measures, it is reasonable to guess that the continuous dynamics (2) approximate the particle dynamics (USA) in the limit of an infinite number of particles, at least in a certain timescale. In the following section, we show a propagation of chaos result, which puts this consideration on a rigorous footing and is a prelude to our results on dynamical metastability.

---

[1] Which has the advantage of having a gradient flow structure.

[2] For these specific (non-absolutely continuous with respect to Lebesgue) measures $\mu$, this equation needs to be interpreted with the weak formulation. Further, note that div here and throughout indicates the divergence intrinsic to $\mathbb{S}^{d-1}$, i.e. the covariant divergence induced by the standard metric on $\mathbb{S}^{d-1}$.

## 3 PROPAGATION OF CHAOS RESULTS AND MEAN FIELD LIMIT

In this section, we establish the rigorous statements for the mean field limit of models (USA) and (SA). In what follows, we denote by $W_1(\cdot, \cdot)$ the 1-Wasserstein distance[3] on $\mathcal{P}(\mathbb{S}^{d-1})$ (the space of probability measures on $\mathbb{S}^{d-1}$).

Consider a cloud of $N$ points on $\mathbb{S}^{d-1}$, $\Xi^{(N)} := \left(x_1^{(N)}, \ldots, x_N^{(N)}\right) \in \left(\mathbb{S}^{d-1}\right)^N$ which we think of as initial conditions for either (SA) of (USA). Let the associated evolution be $\Xi^{(N)}(t) := \left(x_1^{(N)}(t), \ldots, x_N^{(N)}(t)\right)$, and consider its empirical measure

$$\mu_{\Xi^{(N)}}(t) := \frac{1}{N} \sum_{j=1}^{N} \delta_{x_j^{(N)}(t)}.$$

The following theorem proves that the convergence of a sequence of empirical measures $\Xi^{(N)}$ to a limiting one $\mu_0$ (e.g., if $\{x_j\}$ are drawn *iid* from $\mu_0$) is preserved by the flow of the PDE for any finite time $t$, i.e., that the $N$-particles measure remains close to the dynamics of the particles' law as described by the PDE.

**Theorem 3.1** (Mean field limit). *Assume that there exists $\mu_0 \in \mathcal{P}(\mathbb{S}^{d-1})$ such that $W_1(\mu_{\Xi^{(N)}}(0), \mu_0) \to 0$ as $N \to \infty$. Let $\mu(t)$ be the unique weak solution of the associated mean field dynamics (2) with initial condition $\mu(0) = \mu_0$. Then, for any fixed $t \geq 0$, as $N \to \infty$,*
$$W_1(\mu_{\Xi^{(N)}}(t), \mu(t)) \to 0.$$

**Remark 3.2.** *Global existence of weak solutions to (2) in $\mathcal{P}(\mathbb{S}^{d-1})$ is classical, and follows from a-priori estimates on the kernel $\chi[\mu]$.*

The proof of the above theorem is provided for completeness in Appendix A. Following classical references (Sznitman, 1991), we prove the desired estimate by propagating the error estimate at time $t = 0$ to any given positive time by using an exponential (in $t$) bound on the rate of separation of trajectories, also called Dobrushin's bound (Dobrushin, 1979), summarized in A.4.

To conclude this section, we highlight the fact that the estimate presented in the above theorem only holds on finite time intervals. This is a result of the exponential degeneration in time of Dobrushin's bound, which is expected to hold in general (as it is a form of Cauchy stability). This degeneration is indeed inevitable for the system at hand (i.e., this system does not obey uniform in time propagation of chaos) as proven in a counterexample in Appendix B.

As we shall see in the upcoming section, despite the above exponential degeneration, estimates on longer time intervals may still be established, providing insight on the qualitative behavior of the system beyond the finite-time horizon. This is a necessary step to characterize the emergence of meta-stable phases, i.e., states existing for long – in $N$ – time intervals.

## 4 DYNAMIC META-STABILITY RESULTS

In (Geshkovski et al., 2023), it is observed that the tokens $x_i(t)$ in models (SA) and (USA) exhibit exponential convergence to a single clustered configuration in certain regimes, as time goes to infinity. In an intermediate time range, however, the authors observe the existence of long-time meta-stable states. Their simulations reveal a two-phase dynamic: an initial phase characterized by the formation of multiple clusters, followed by a pairwise merging of these clusters, ultimately leading to a single point mass distribution. In this section, we rigorously describe both the onset and the development of such dynamical metastability phenomenon.

### 4.1 META-STABILITY: SETUP AND DISCUSSION

We restrict our attention to a class of models which arise as a generalization of (USA):

$$\begin{cases} \partial_t \mu_t + \operatorname{div}(\mu_t \nabla(W * \mu_t)) = 0 & \text{for } (t, x) \in \mathbb{R}_{\geq 0} \times \mathbb{S}^{d-1}, \\ \mu_{|t=0} = \mu_0 & \text{for } x \in \mathbb{S}^{d-1}. \end{cases} \quad (3)$$

---

[3]See Appendix A.1 the precise definition.

Here, the convolution operator $*$ is defined canonically as follows[4]:

$$(f * g)(x) = \int_{\mathbb{S}^{d-1}} f(\langle x, y \rangle) g(y) \mathrm{d}\sigma_{\mathbb{S}^{d-1}}(y), \tag{4}$$

where $f : [-1, 1] \to \mathbb{R}$, $g : \mathbb{S}^{d-1} \to \mathbb{R}$ and $\mathrm{d}\sigma_{\mathbb{S}^{d-1}}$ is the standard Lebesgue measure on $\mathbb{S}^{d-1}$. In addition div and $\nabla$ are the intrinsic divergence and gradient in $\mathbb{S}^{d-1}$, induced by the standard metric on $\mathbb{S}^{d-1}$.

Note that, setting $W(q) := \beta^{-1} \exp(\beta q)$ we recover the case of the (mean-field) dynamics (USA-MF). However, in the spirit of keeping our discussion general, we analyze the above equation for more general kernels $W$, making the following assumptions about its properties.

**Assumption 1.** *$W$ is a smooth ($C^\infty$) function on $[-1, 1]$.*

**Assumption 2.** *Let $\hat{W}_k$ be the $k-$th Gegenbauer coefficient of $W$ (see (C.1.2) for the definition), and $\gamma_k := k(k + d - 2)\hat{W}_k$. The sequence $\{\gamma_k\}_{k \geq 0}$ has a finite maximum $\gamma_{max} > 0$. Moreover, this maximum is attained only at a single value $k_{max} > 0$.*

**Remark 4.1.** *Note that Assumption 1 is trivially satisfied in the transformers model. Furthermore, for $d = 2$ the coefficients $\hat{W}_k$ are simply given by $\beta^{-1} I_k(\beta)$, where $I_k$ is the $k$-th order modified Bessel funcion of the first kind. Classical asymptotic estimates on Bessel functions (Abramowitz & Stegun, 1948, p. 360, 9.1.10) show that the maximum $\gamma_{max}$ is finite for all $\beta$ and it is non-unique only on a set of Lebesgue measure $0$. Furthermore, as $\beta \to \infty$, $k_{max} \approx \sqrt{\beta}$.*

Assumption 2 is related to the concept of H-stability introduced by (Ruelle, 1999) and, as discussed in (Carrillo et al., 2020) for the torus, determines the instability of the uniform measure and the properties of phase transitions in the presence of noise. We will in particular show that $\hat{W}_{k_{max}}$ also controls the time scale and the type of emerging symmetries.

In what follows, motivated by the absence of a preferential direction in embedding space, we consider as the initial condition the empirical measure associated with $N$ tokens sampled independently and uniformly on $\mathbb{S}^{d-1}$. Although there is an extensive body of work on fluctuations of the mean-field limit ((Fernandez & Méléard, 1997; Lancellotti, 2009)), these studies generally focus on a finite time interval $[0, T]$, which is insufficient for observing the meta-stable phase. Indeed, in our setting, the size of the perturbation scales as $O(N^{-1/2})$, vanishing at any finite time $T$ in the $N \to \infty$ limit. To observe the effect of fluctuations at initialization at macroscopic scales, a much more refined analysis is required. To carry out such analysis, we decompose the meta-stable phase in three parts: the *linear phase*, the *quasi-linear phase*, and the *clustering phase*, as depicted in Figure 1:

- **Linear phase.** Starting from a very small neighborhood of the uniform measure, the perturbation coalesces towards the dominant mode determined by $k_{max}$ until it reaches roughly size $\epsilon \sim O(N^{-1/4})$. We describe the dynamics in this phase in a precise fashion, providing quantitative estimates for the distance between the solution to the nonlinear PDE in (3) and its linearization around the uniform measure.

- **Quasi-linear phase.** After exiting the ball of radius $\epsilon \sim O(N^{-1/4})$, the perturbation remains close to the nonlinear evolution of the dominant mode sufficiently long so that its size exceeds a small threshold $\delta > 0$ independent of the number of particles. Moreover, when exiting the quasi-linear phase, the solution is arbitrarily close (as $N \to \infty$) to the invariant manifold selected by $k_{max}$. This manifold is characterized by measures invariant under $\frac{2\pi}{k_{max}}$ rotations for $d = 2$. For $d \geq 3$, the description of the above manifold is provided in Section 4.4.

- **Nonlinear and collapsing phase.** This phase consists of any finite-time interval after the quasi-linear phase. In this phase, the solution preserves the structure (e.g., periodicity) that emerged during the preceding phases. The exact evolution in this phase depends on the specific form of the interaction potential $W$. In the case of Transformers, for $W(q) = \beta^{-1} e^{\beta q}$, numerical simulations show that, after exiting the quasi-linear phase, the dominant mode collapses into $k_{max}$ clusters.

---

[4]We recall the definition for integrable functions, however it is standard to extend this to the convolution of a smooth $W$ with a measure $\mu$.

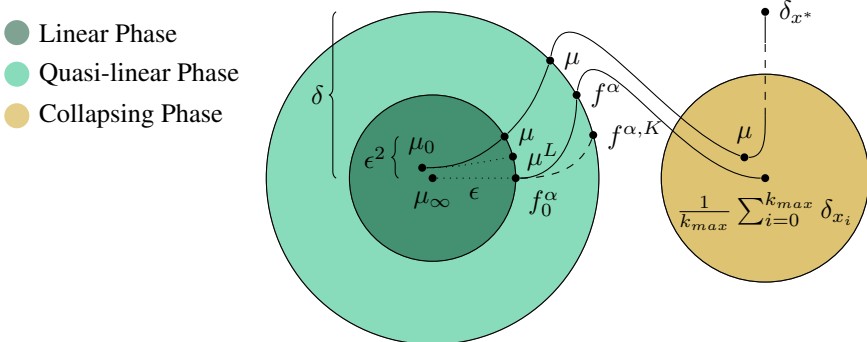

Figure 1: Schematic representation of our decomposition of the dynamics. Here, $\mu$ represents the "true" evolution of the system, while $\mu_L$ represents the linearized evolution around the uniform measure $\mu_\infty$; $f^\alpha$ denotes the evolution of the mean-field PDE with initial conditions $f_0^\alpha$ on the invariant manifold selected by $k_{max}$, and $f^{\alpha,K}$ is the approximation by Grenier's iterative scheme.

After the three phases described above, a final phase occurs, which we refer to as the *late-time phase*. During this phase, as shown in (Markdahl et al., 2017) and (Criscitiello et al., 2024), the multiple clusters eventually converge to a single point, possibly going through several meta-stable phases with an intermediate number of clusters.

## 4.2 META-STABILITY: PRECISE STATEMENTS

To avoid technical complications and to improve clarity of exposition, we focus on the system (3) in dimension $d = 2$, identifying $\mathbb{S}^1$ with $[0, 2\pi)$. The analysis conducted here for the linear and quasi-linear phase can be immediately generalized to the case $d > 2$ as we outline in Section 4.4.

We start by recalling our setup. We draw $N$ tokens $x_i(0)$ at initialization, independently and uniformly at random on $\mathbb{S}^{d-1}$, and consider the resulting (random) evolution under (USA). Let $\mu_t$ be the empirical measure associated to the tokens $\{x_i(t)\}_{i=1,\dots,N}$, and let $\mu_0$ be $\mu$ at time $t = 0$. We define the perturbation $\rho_t := \mu_t - \mu_\infty$, omitting its dependence on $N$ for clarity, and the corresponding characteristic time for the linear phase by:

$$T_1 := \frac{1}{\gamma_{max}} \ln \left( \frac{N^{-1/4}}{\|\rho_0\|_{H^{-1}}} \right).$$

where here and throughout $\|\cdot\|_{H^{-1}}$ denotes the Sobolev norm with $p = 2$ and negative exponent $k = -1$ defined in (14). Moreover, $T_1$ is well-defined and grows logarithmically in $N$, as a consequence of the the Central Limit Theorem (see also Lemma E.3).

**Theorem 4.2** (Linear phase). *The measure $\mu(T_1)$ can be decomposed as:*

$$\mu(T_1) = \mu_\infty + N^{-1/4} \frac{(\hat{\rho}_0)_{k_{max}}}{\|\rho_0\|_{H^{-1}}} \cos(k_{max}\theta) + R, \tag{5}$$

*where $R$ is a remainder which satisfies $\|R\|_{H^{-1}} = o(N^{-1/4})$ in probability, and $(\hat{\rho}_0)_{k_{max}}$ denotes the Fourier coefficient of $\rho_0$ with index $k_{max}$. Thus, as $N \to \infty$, the evolution at time $T_1$ is $k_{max}$ periodic and has typical size $N^{-1/4}$ away from the uniform measure.*

From $T_1$ onwards, the dynamics are well-approximated by the mean-field PDE with initial data at $T_1$:

$$\begin{cases} \partial_t f^\alpha = -\partial_\theta \left( f^\alpha \nabla W * f^\alpha \right) \\ f_0^\alpha(0) = \mu_\infty + \alpha \cos(k_{max}\theta), \end{cases} \tag{6}$$

where $\alpha = N^{-1/4} \frac{(\hat{\rho}_0)_{k_{max}}}{\|\rho_0\|_{H^{-1}}}$. Importantly, since the first equation in display (6) is invariant under rotation (i.e., translation in $\theta$), it will preserve the periodicity (if any) of its initial condition. Consequently, we note that the function $f^\alpha$ will be $k_{max}$-periodic for all times.

Let $\delta > 0$ be a fixed small parameter, independent of $N$. As the following theorem shows, the characteristic time to reach the end of the quasi-linear phase is given by:

$$T_2 := \frac{1}{\gamma_{\max}} \ln(\delta/\alpha). \tag{7}$$

**Theorem 4.3** (Quasi-linear phase). *Let $f^\alpha$ be the solution to the initial value problem (6) and $\delta > 0$ small enough. Then, we have*

$$W_1(\mu(T_1 + T_2), f^\alpha(T_2)) \to 0,$$

*in probability as $N \to \infty$ and*

$$W_1(\mu(T_1 + T_2), \mu_\infty) > \delta.$$

**Remark 4.4.** *Recalling that $f^\alpha$ has periodicity $k_{max}$, Theorem 4.3 states that the evolution in the mean field limit, when exiting the quasi-linear phase, converges to a $k_{max}$-periodic function, while being a fixed amount $\delta$ away from the equilibrium measure. In other words, this result proves that when exiting the quasi-linear phase, $\mu$ (approximately) displays a very specific structure.*

Finally, after the quasi-linear phase, we establish the mean-field limit in the clustering phase. To do so we require the following assumption:

**Assumption 3.** *Let $f_* := \lim_{\alpha \to 0} f^\alpha(T_2)$, and consider the initial value problem (6) with initial data $f_*$ imposed at time $T_2$. Then, as $t \to \infty$,*

$$W_1(f_*(t), \mu_{cluster}) \to 0 \qquad where \qquad \mu_{cluster} := \frac{1}{k_{max}} \sum_{j=0}^{k_{max}-1} \delta_{\frac{2\pi j}{k_{max}}}. \tag{8}$$

Let us motivate Assumption 3. Indeed, it is relatively straightforward (using the gradient flow structure) to show that, as time goes to infinity, $f_*(t)$ converges (in the weak sense) to a sum of a number of delta masses located at several points (see Lemma E.5). Moreover, again by translation invariance in $\theta$, the limit will be $k_{max}$ periodic. Thus, Assumption 3 is true if we only require the limit to be a $k_{max}$ periodic superposition of a number of delta masses.

To motivate why $\mu_{\text{cluster}}$ should have the specific form (8), let us focus our attention on the dynamics of equation (6) on each interval of periodicity (for simplicity, consider the interval $I := [-\pi/k_{\max}, \pi/k_{\max}]$). Following an analogous reasoning as in (Markdahl et al., 2017) it can be shown that the evolution under the dynamics of (6) (restricted to $I$ via periodicity) of almost every empirical measure will tend, as time goes to infinity, to a single delta function. Thus, Assumption 3 holds for almost every empirical measure, which leads us to believe that it should hold for the particular $f_*$ under consideration.

Under Assumption 3, we obtain the following statement.

**Theorem 4.5** (Clustering phase). *Let $T_3 > 0$ be a finite time. As $N$ tends to infinity, $\mu_{(T_1+T_2+T_3)}$ is approximated by $f^\alpha(T_2+T_3)$ in Wasserstein 1-distance, with an error vanishing in probability as $N$ tends to infinity. In particular, provided that Assumption 3 holds, and picking the time $T_3$ sufficiently large, as $N$ goes to infinity, $\mu_{(T_1+T_2+T_3)}$ is arbitrarily close (in the Wasserstein 1-distance and in probability) to the measure $\mu_{cluster}$ defined in display (8).*

**Remark 4.6.** *Note that, without requiring assumption 3, an analogous rigorous statement can be obtained in which the limiting measure is a $k_{max}$ periodic superposition of delta masses. However, in this case a number $k \geq k_{max}$ of clusters, possibly of different intensity, may emerge.*

### 4.3 DESCRIPTION OF THE PROOF

The technical machinery which enables us to achieve the proof of the above Theorems is contained in Appendix C. First, we introduce key technical definitions and statements in Section C.1, and the linearization of our problem in Section C.2.

**Linear phase.** Section C.3 is devoted to the proof of Proposition C.10, the quantitative statement of Theorem 4.2. First, we improve, via Lemma C.7, the Lyapunov exponent on times smaller than $T_1$, by studying explicitly the linearized operator around $\mu_\infty$. This Lemma employs a duality argument

to avoid derivative loss, and allows us to show a quantitative improvement of Dobrushin's estimate, which in turn is enough to show propagation of chaos for times comparable to $T_1$. These ingredients are enough to conclude the proof of Proposition C.10.

**Quasi-linear phase.** In Section C.4, we construct a genuinely nonlinear solution which arises from the unstable mode $\cos(k_{max}\theta)$ after time $T_1$. The main technical issue in this regard is that we only have a poor estimate of the nonlinear growth rate $C > 0$ of solutions to the mean-field equation (appearing in Dobrushin's estimate). In particular, a direct approach would work if $2\gamma_{max} > C$ (i.e., the nonlinear growth rate is slower than twice the growth rate of the unstable mode) for the relevant times $t < T_1 + T_2$, but this estimate does not seem to be available. By employing Grenier's scheme (Grenier, 2000), we consider a higher-order approximation to the dynamics, which allows us to relax the above condition to $nk_{max} > C$, where $n > 1$ is the order of our approximation. This reasoning allows us to prove Proposition C.15 on the quasi-linear phase.

**Collapse phase.** At time $T_2$, as $N \to \infty$, the evolution concentrates around the $k_{max}$ periodic function $f_*$ considered in assumption 3. We then use Dobrushin's estimates, together with the specific form of the infinite time limit provided by assumption 3 to conclude the statement of Theorem C.23. Note in particular that, after time $T_2$, the time required for clustering to occur does not depend on $N$. Consequently, Dobrushin's estimates apply directly here.

## 4.4 META-STABLE PHASE IN HIGHER DIMENSIONS

In higher dimensions, a similar analysis of the meta-stable phase can be carried out as in the lower-dimensional case (see D for details). The definitions and arguments from the previous sections naturally extend to the case $d \geq 2$ due to the properties of spherical harmonics and Gegenbauer coefficients. The uniform measure remains an unstable equilibrium. The key difference is that the dominant mode $f^\alpha$, which emerges in the linear phase as described in Theorem 4.2, is now a superposition of spherical harmonics of degree $k_{max}$, solving:

$$\begin{cases} \partial_t f^\alpha = -\partial_\theta \left( f^\alpha \nabla W * f^\alpha \right) \\ f_0^\alpha(T_1) = \mu_\infty + \sum_{j=0}^{Z_{k_{max}}^d} \alpha_j Y_{k_{max},j}, \end{cases} \tag{9}$$

where $Z_{k_{max}}^d$ is the multiplicity of spherical harmonics of degree $k_{max}$ in dimension $d$.

This function belongs to the subspace $\mathcal{H}_{k_{max}}$, invariant for the continuity equation (3), defined as:

$$\mathcal{H}_k := \{\mu \in \mathcal{P}(\mathbb{S}^{d-1}) : \langle \mu, Y_{l,j} \rangle = 0 \quad \forall l \text{ not divisible by } k\},$$

that generalizes the invariance under $\frac{2\pi}{k_{max}}$-rotations. Hence, Theorem 4.3 can be reformulated in this higher-dimensional context as:

**Theorem 4.7** (Quasi-linear phase). *Let $f^\alpha$ be the solution to the initial value problem (9) and $\delta > 0$ small enough. Then, we have*

$$W_1(\mu(T_1 + T_2), f^\alpha(T_2)) \to 0,$$

*in probability as $N \to \infty$ and*

$$W_1(\mu(T_1 + T_2), \mu_\infty) > \delta.$$

Again, Theorem 4.7 states that the evolution in the mean field limit, when exiting the quasi-linear phase, converges to a function in $\mathcal{H}_{k_{max}}$, while being a fixed amount $\delta$ away from the equilibrium measure. This proves that the measure displays a very specific structure at the end of the quasi-linear phase. A natural extension of the analysis is to characterize the stable equilibria of the dynamics when restricted to the subspace $\mathcal{H}_{k_{max}}$, excluding the uniform measure. These equilibria must be finite unions of submanifolds of dimension at most $d - 2$ (see Lemma E.5). Furthermore, if the measure is the empirical measure representing a set of points, being in $\mathcal{H}_{k_{max}}$ (Bannai et al. (2015)) implies that these points form a spherical $k_{max} - 1$-design. This condition indicates specific properties regarding the minimum number of clusters, their geometric structure, and the distribution of tokens, which have been extensively explored in the literature and are an active area of research.

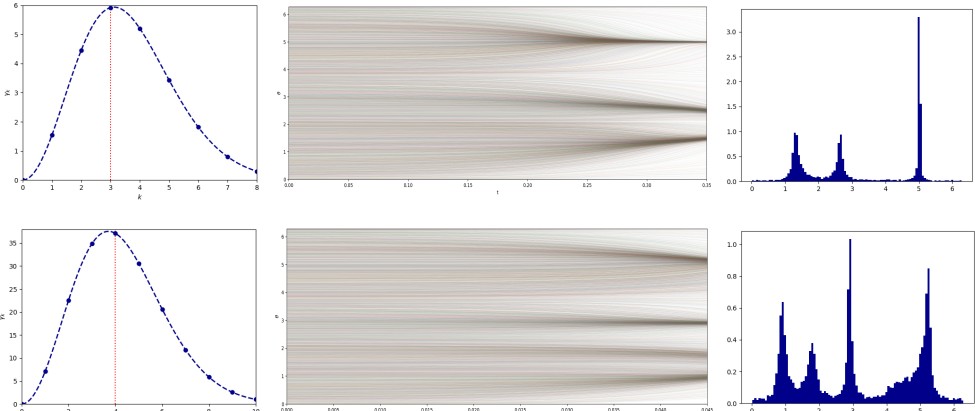

Figure 2: On the left, the plots of $\gamma_k$ as a function of $k$ for $\beta = 5$ (top) and $\beta = 7$ (bottom) are shown. The dashed blue line represents the value of $\gamma_k$ generalized to real $k$. From these graphs, we observe that the corresponding values of $k_{\max}$ are 3 and 4, respectively, representing the predicted number of clusters in the large-$N$ limit. In the center, numerical simulations depict particle trajectories for $\beta = 5$ (top) and $\beta = 7$ (bottom), with $10^4$ particles whose initial conditions are sampled uniformly at random. On the right, the histograms of particle distributions at the end of the simulations are displayed, showcasing the formation of 3 and 4 clusters respectively.

## 5 NUMERICAL EXPERIMENTS

The first experiment aims to illustrate the relationship between the parameter $\beta$ and the number of clusters formed: an immediate manifestation of the symmetry described in Theorem 4.3 and the clustering tendency of the dynamics. Specifically, the left column of Figure 2 illustrates the values of the coefficients $\gamma_k$ for the Transformer model USA with $\beta$ set to 5 and 7. Assumption 2 is satisfied with $k_{max}$ values of 3 and 4, respectively. By Theorem 4.2 and Proposition C.20, these values correspond to the expected number of clusters observed during the meta-stable phase. To confirm this, we simulate in Figure 2 the trajectories of $10^4$ particles on $\mathbb{S}^1$, whose initial conditions are sampled uniformly at random. The angular ODEs system (10), equivalent to Eq. (USA), is numerically solved using the Euler method with a time step $dt = 5 \times 10^{-4}$. Notice the different timescales of the two simulations, the meta-stable phase and the emergence of the predicted 3 and 4 clusters. The computation is performed using PyTorch in double precision on a Nvidia Tesla T4 GPU.

With the second experiment we demonstrate the decomposition of Eq. (5) and the emergence of periodicity in the token distribution. To further investigate the asymptotic limit on the number of tokens, and given the computational constraints on simulating a large number of particles, we chose to examine the limiting case of an absolutely continuous initial condition. Given that the results in Theorem 4.2 apply to general perturbations of uniform measures, we consider as initial condition, $\mu_0$, the uniform measure slightly perturbed by white noise. Figure 4 illustrates the evolution of $\mu_t$ starting from $\mu_0$. The continuity equation (3), more precisely its angular counterpart Eq.(11), is numerically solved using the Lax–Friedrichs method by discretizing the spatial domain into $10^4$ grid points over the interval $[0, 2\pi]$, with a spatial step $dx = 2\pi \times 10^{-4}$ and a time step $dt = 0.05dx$. The resulting dynamics consistently confirm our theoretical expectations.

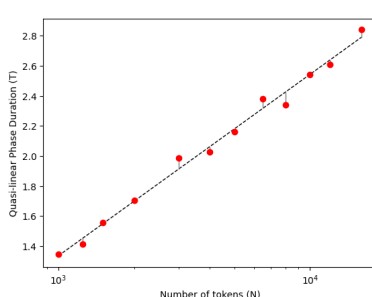

Figure 3: Plot of the average time required for the empirical measure to exceed a fixed threshold. Note the logarithmic scale on the $x$-axis.

In the third experiment, we demonstrate that the duration of the meta-stable phase is $O(\ln(N))$, as predicted by Theorem 4.2 and Theorem 4.3. We fix $\beta = 2$ and run multiple simulations with the same setup of the first experiment, with the number of particles ranging from $N = 1000$ to $N = 16000$. The simulations are terminated when the approximate total variation distance between

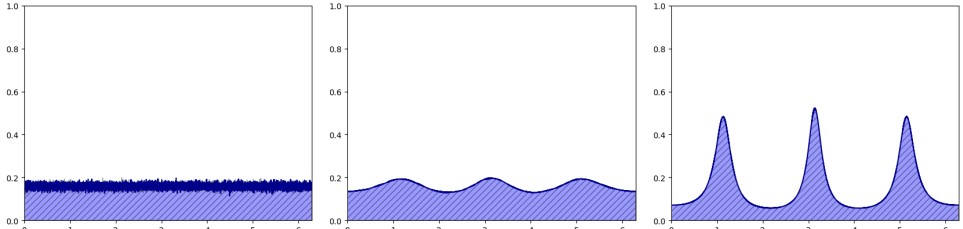

Figure 4: Evolution of the initial condition given by the uniform measure perturbed by white noise ($\sigma = 0.01$) with $\beta = 5$. Note that the emerging solution is 3-periodic as predicted by Theorem 4.3.

the uniform distribution and the token distribution (computed using 100 bins for the histogram) exceeds a fixed threshold. Figure 3 shows the average results from 20 simulations per particle number. We observe a clear logarithmic dependence in $N$ of the exit time from the quasi-linear regime, aligning with the theoretical expectations. Code is available at this GitHub-Repository.

## 6  CONCLUSIONS AND FUTURE WORKS

In this paper we rigorously investigate the dynamics of $N$ tokens, interpreted as exchangeable particles, through the depth of a simple transformer model in the limit of a large $N$. In this limit, the dynamic meta-stability of the system can be captured by studying the emergence of symmetry at initialization through a careful analysis of the evolving fluctuations in the systems. In particular, we show that, after a time that depends logarithmically on $N$, the system exits from a ball containing the uniform measure displaying, approximately, a specific structure (a certain periodicity in the case $d = 2$) that depends explicitly on the inverse temperature parameter $\beta$ and on the dimension $d$. Since the equation describing the dynamics of this model preserves such structure, this observation allows to capture the emergence of dynamically meta-stable solutions and to characterize them explicitly. In practice, this translates in a detailed understanding of the representations produced by the network at finite but large depths, and how these representations vary with the depth as a function of $N$.

While $10^4$ tokens might seem a large number, many modern models can handle significantly larger context lengths. For instance, ChatGPT can process up to 128k tokens, and Gemini 1.5 Pro supports up to 2 million tokens. As the context lengths and the depth of large language models continue to expand, the large-$N$ limit and the meta-stable effects discussed here will become increasingly relevant. Indeed, we demonstrated the existence of a strong interconnection among the main hyper-parameters of a Transformer model: number of tokens, temperature, embedding dimension, and model depth. These hyper-parameters cannot be adjusted independently of one another.

Admittedly, our work makes strong assumptions on the structure of the model, e.g., absence of MLP layer and $Q = K = V = Id$. The latter assumption can be immediately relaxed to $Q^T K = \lambda Id$, and quite possibly to $Q^T K = V$, as the gradient flow structure is preserved in that case. While these assumptions are still far from realistic, numerical experiments in (Geshkovski et al., 2023) show that BERT displays qualitatively similar token clustering behavior to the one of our model. On the other hand, the choice of MLP layer will have a dramatic impact on the dynamical landscape (Cowsik et al., 2024). The qualitative effect of this extra term on the dynamics depends heavily on the specific choice of MLP weights, but a linearized analysis similar to ours is expected to go through in a neighborhood of the (new) unstable equilibria of the model. Despite these limitations, we believe isolating properties of the full model in simplified settings such as the one considered here is a crucial step toward building a comprehensive understanding of real-world transformers. Combining this result with complementary ones on MLP and nontrivial $Q, K, V$ is left for future research.

This work hints at a number of further interesting questions. First, an interesting avenue of future research consists of proving that Assumption 3 holds, i.e., that the solution to the PDE (2) as it exits the quasi-linear phase indeed converges to a certain number of delta measures. This result goes well beyond the ones presented in (Cohn & Kumar, 2007), as it requires characterising the solution of the PDE with a specific initial condition that may fall in the set of measure $0$ in the previously established result. It would also be interesting to investigate the effect of adding noise - motivated by considering randomly initialized linear layers between attention layers before training - to the ODEs (SA) and (USA) and, as a consequence, on the emergence of symmetry and clustering in the resulting system.

ACKNOWLEDGMENTS

We thank Philippe Rigollet for discussions that motivated this work. AA and GB thank the Mathematics Department at the University of Pisa, where part of this work was carried out. AA acknowledges partial support of Dipartimento di Eccellenza, UNIPI, the Future of Artificial Intelligence Research (FAIR) foundation (WP2), PRIN 2022 project ConStRAINeD (2022XRWY7W), and PRA Project APRISE.

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

# Appendix

## A  PROPAGATION OF CHAOS

### A.1  PRELIMINARIES: WASSERSTEIN DISTANCE

Let $(X, d)$ be a compact metric space and let $\mathcal{P}(X)$ be the set of probability measures on $X$. The $p-$Wasserstein distance is defined as:

$$W_p(\mu, \nu) := \left( \min_{\gamma \in \Pi(\mu, \nu)} \int d(x, y)^p d\gamma(x, y) \right)^{1/p}.$$

It is a well defined distance and in compact spaces satisfies for all $p \geq 1$:

$$W_1(\mu, \nu) \leq W_p(\mu, \nu) \leq \text{diam}(X)^{\frac{p-1}{p}} W_1(\mu, \nu)^{\frac{1}{p}}.$$

Moreover, under the compactness assumption, for $p \in [1, +\infty[$, we have $\mu_n \rightharpoonup \mu$ if and only if $W_p(\mu_n, \mu) \to 0$.

Another common characterization of the distance $W_1$ is a consequence of the duality theorem of Kantorovich and Rubinstein: when $\mu$ and $\nu$ have bounded support,

$$W_1(\mu, \nu) = \sup \left\{ \int_X f(x) d(\mu - \nu)(x) \big| f : X \to \mathbb{R} \text{ continuous }, \text{Lip}(f) \leq 1 \right\}.$$

For other properties and results related to the Wasserstein distance, the reader can refer to (Ambrosio et al., 2008; Panaretos & Zemel, 2020).

### A.2  ANGULAR EQUATION AND MEAN-FIELD PDE CONVERGENCE

In this section we provide a proof of the Dobrushin's estimate for the Transformers model. For clarity we restrict our attention to the case where $d = 2$ for the model USA. Nevertheless, the same arguments can be extended to prove the estimate for $d \geq 3$ and for the model SA.

An interacting particle system on $\mathbb{S}^1$ can be equivalently described by associating the particles $x_i(t) \in \mathbb{S}^1$ with the corresponding angles $\theta_i(t) \in \mathbb{T}$. The angular representation of the system of ODEs for the model USA can be expressed as follow:

$$\dot{\theta}_i = -\frac{1}{N} \sum_{j=1}^{N} e^{\beta \cos(\theta_i - \theta_j)} \sin(\theta_i - \theta_j). \tag{10}$$

Then the empirical measure of the angles, $\nu(t) = \frac{1}{N} \sum_{j=1}^{N} \delta_{\theta_j(t)}$, is a solution of the continuity equation:

$$\partial_t \nu(t) + \partial_\theta(\chi[\nu(t)]\nu(t)) = 0, \tag{11}$$

where:

$$\chi[\nu](\theta) = \int h_\beta'(\theta - \omega) \, d\nu(\omega)$$

and $h_\beta(\theta) = \frac{1}{\beta} e^{\beta \cos(\theta)}$. With this notation and these definitions, Dobrushin's estimate can be formulated as follows:

**Theorem A.1.** *Let $\mu_t$, $\nu_t$ be two solutions of Eq. (11), with initial conditions respectively $\mu_0, \nu_0 \in \mathcal{P}(\mathbb{S}^1)$. Then $\forall t \geq 0$:*

$$W_1(\mu_t, \nu_t) \leq e^{2Ct} W_1(\mu_0, \nu_0),$$

*with $C = \|h_\beta''\|_{L^\infty}$.*

We denote $T_t[\mu](x)$ the flow of our system with initial condition $x \in \mathbb{S}^1$ and vector field $\chi[\mu_t]$. Furthermore, given a generic measurable map $f$, we denote the push-forward of the measure $\mu$ under the map $f$ as $f\#\mu$. In particular Eq. (11) is equivalent to $\nu_t = T_t[\nu]\#\nu_0$.

The proof of Theorem A.1 is a consequence of the Lipschtzianity of the vector field $\chi[\mu_t]$ and of the flow $T_t[\mu]$, which are shown in the next two lemmas:

**Lemma A.2.** *Let $\mu, \nu \in \mathcal{P}(\mathbb{S}^1)$, then:*

$$\|\chi[\mu] - \chi[\nu]\|_{L^\infty} \leq \|h_\beta''\|_{L^\infty} W_1(\mu, \nu).$$

*Proof.* By definition of $\chi$:

$$\|\chi[\mu] - \chi[\nu]\|_{L^\infty} = \sup_{\theta \in \mathbb{S}^1} \left| \int h_\beta'(\theta - \omega)[d\mu(\omega) - d\nu(\omega)] \right| \leq$$

$$\leq \|h_\beta''\|_{L^\infty} \sup_{\theta \in \mathbb{S}^1} \left| \int \frac{h_\beta'(\theta - \omega)}{\|h_\beta''\|_{L^\infty}} [d\mu(\omega) - d\nu(\omega)] \right|.$$

Since $\frac{h_\beta'(\theta - \omega)}{\|h_\beta''\|_{L^\infty}}$ is 1-Lipschitz we can conclude by definition of $W_1$. $\qquad\square$

**Lemma A.3.** *Let $\mu \in \mathcal{P}(\mathbb{S}^1)$, then:*

$$\|\chi[\mu]\|_{Lip} \leq C, \quad \|T_t[\mu]\|_{Lip} \leq e^{Ct}.$$

*where $C = \|h_\beta''\|_{L^\infty}$.*

*Proof.* For the first part:

$$\|\chi[\mu](\theta_1) - \chi[\mu](\theta_2)\| \leq \int |h_\beta'(\theta_1 - \omega) - h_\beta'(\theta_2 - \omega)| d\mu(\omega) \leq \|h_\beta''\|_{L^\infty}(\theta_1 - \theta_2).$$

For the second part:

$$\frac{d}{dt} |T_t[\mu](\theta_1) - T_t[\mu](\theta_2)| \leq \left| \frac{d}{dt} T_t[\mu](\theta_1) - \frac{d}{dt} T_t[\mu](\theta_2) \right| =$$

$$= |\chi[\mu_t](T_t(\theta_1)) - \chi[\mu_t](T_t(\theta_2))| \leq C |T_t(\theta_1) - T_t(\theta_2)|.$$

The conclusion follows as a consequence of Gronwall's lemma. $\qquad\square$

Now we can return to the theorem:

*Proof of Theorem A.1.*

$$\begin{aligned} W_1(\mu_t, \nu_t) = W_1(T_t[\mu]\#\mu_0, T_t[\nu]\#\nu_0) \leq \\ \leq W_1(T_t[\mu]\#\mu_0, T_t[\mu]\#\nu_0) + W_1(T_t[\mu]\#\nu_0, T_t[\nu]\#\nu_0) \end{aligned} \tag{12}$$

For the first term:

$$W_1(T_t[\mu]\#\mu_0, T_t[\mu]\#\nu_0) = \sup_{\|\phi\|_{Lip} \leq 1} \int_{\mathbb{S}^1} \phi \, d[T_t[\mu]\#\mu_0 - T_t[\mu]\#\nu_0] =$$

$$= \sup_{\|\phi\|_{Lip} \leq 1} \int_{\mathbb{S}^1} (\phi \circ T_t[\mu]) \, d[\mu_0 - \nu_0] =$$

$$= e^{Ct} \sup_{\|\phi\|_{Lip} \leq 1} \int_{\mathbb{S}^1} (e^{-Ct} \phi \circ T_t[\mu]) \, d[\mu_0 - \nu_0] \leq$$

$$\leq e^{Ct} \sup_{\|\psi\|_{Lip} \leq 1} \int_{\mathbb{S}^1} \psi \, d[\mu_0 - \nu_0] = e^{Ct} W_1(\mu_0, \nu_0),$$

where the last row follow from Lemma (A.3).

For the second term:

$$W_1(T_t[\mu]\#\nu_0, T_t[\nu]\#\nu_0) = \sup_{\|\phi\|_{Lip}\leq 1} \int_{\mathbb{S}^1} (\phi \circ T_t[\mu] - \phi \circ T_t[\nu]) \ d\nu_0 \leq$$

$$\leq \int_{\mathbb{S}^1} |T_t[\mu] - T_t[\nu]| \ d\nu_0 := \lambda(t).$$

Now,

$$\frac{d\lambda(t)}{dt} \leq \int \left| \frac{d}{dt} T_t[\mu] - \frac{d}{dt} T_t[\nu] \right| d\nu_0 =$$

$$= \int |\chi[\mu_t] \circ T_t[\mu] - \chi[\nu_t] \circ T_t[\nu]| \ d\nu_0 \leq$$

$$\leq \int |\chi[\mu_t] \circ T_t[\mu] - \chi[\mu_t] \circ T_t[\nu]| \ d\nu_0 + \int |\chi[\mu_t] \circ T_t[\nu] - \chi[\nu_t] \circ T_t[\nu]| \ d\nu_0.$$

By Lemma (A.3) for the first term, and by the properties of $T_t$ for the second term (in particular $\mu_t = T_t[\mu]\#\mu_0$), we get:

$$\frac{d\lambda(t)}{dt} \leq C \int |T_t[\mu] - T_t[\nu]| \ d\nu_0 + \int |\chi[\mu_t] - \chi[\nu_t]| \ d\nu_t \leq$$

$$\leq C\lambda(t) + \|\chi[\mu_t] - \chi[\nu_t]\|_{L^\infty} \leq C[\lambda(t) + W_1(\mu_t, \nu_t)].$$

where the last step follows by Lemma (A.2). As a consequence of Gronwall's inequality it holds:

$$\lambda(t) \leq C \int_0^t e^{C(t-\tau)} W_1(\mu_\tau, \nu_\tau) d\tau.$$

Going back to Equation (12) we get:

$$W_1(\mu_t, \nu_t) \leq e^{Ct} W_1(\mu_0, \nu_0) + C \int_0^t e^{C(t-\tau)} W_1(\mu_\tau, \nu_\tau) d\tau.$$

And to conclude we just need to apply another time Gronwall's lemma:

$$W_1(\mu_t, \nu_t) \leq e^{2Ct} W_1(\mu_0, \nu_0).$$

$\square$

An analogous reasoning in the higher dimensional case, for the dynamics (SA-MF) and (USA-MF) yields the following more general result.

**Theorem A.4** (Dobrushin's estimate in $d$ dimensions). *There exists a constant $C > 0$ depending on $\beta$ such that the following holds. Suppose that $\mu_t, \nu_t$ are two solutions of (2) in the class $C(\mathbb{R}_{\geq 0}, \mathcal{P}(\mathbb{S}^{d-1}))$, with initial conditions respectively $\mu_0, \nu_0 \in \mathcal{P}(\mathbb{S}^{d-1})$. Then $\forall t \geq 0$ :*

$$W_1(\mu_t, \nu_t) \leq e^{2Ct} W_1(\mu_0, \nu_0).$$

*Moreover, $C$ can be computed explicitly. In the case of (SA-MF),*

$$C = \|\nabla_x K_\beta(x, y)\|_{L^\infty(\mathbb{S}^{d-1}\times\mathbb{S}^{d-1})}, \qquad where \qquad K_\beta(x, y) := e^{\beta\langle x, y\rangle} P_x(y).$$

Finally, Theorem 3.1 is a direct consequence of Theorem A.4 and we omit the proof here.

# B ON THE EXPONENTIAL IN TIME DEPENDENCE

In this section, we demonstrate through a straightforward counterexample that the exponential time-dependence in Dobrushin's estimate for the Transformers model cannot generally be improved. The time control is not only non-uniform but must also exhibit exponential growth.

Let's consider for simplicity the case $\beta = 1$ and $N = 2$. The ordinary differential equations system obtained from Eq. (SA) is:

$$\begin{cases} \dot{x}_1 &= P_{x_1}\left(\frac{1}{Z_{\beta,1}}(e^{\langle x_1, x_1\rangle} x_1 + e^{\langle x_1, x_2\rangle} x_2)\right) \\ \dot{x}_2 &= P_{x_2}\left(\frac{1}{Z_{\beta,2}}(e^{\langle x_2, x_1\rangle} x_1 + e^{\langle x_2, x_2\rangle} x_2)\right). \end{cases}$$

It's easier to examine the angle version of the previous ODE system:

$$\begin{cases} \dot{\theta}_1 = & -\frac{1}{Z_{\beta,1}} e^{\cos(\theta_1 - \theta_2)} \sin(\theta_1 - \theta_2) \\ \dot{\theta}_2 = & -\frac{1}{Z_{\beta,2}} e^{\cos(\theta_2 - \theta_1)} \sin(\theta_2 - \theta_1). \end{cases}$$

Using that $Z_{\beta,1} = Z_{\beta,2} =: Z$, then $\omega := \theta_2 - \theta_1$ satisfies:

$$\dot{\omega} = -2 \frac{e^{\cos(\omega)}}{Z} \sin(\omega).$$

Since $\omega(t) = 0$ and $\omega(t) = \pi$ are constant solutions, then for every $\omega_0 \in [0, \pi]$ the corresponding trajectory $\omega(t)$ is confined in $[0, \pi]$ $\forall t \geq 0$, and this implies $\sin(\omega(t)) \geq 0$. Moreover, $\frac{1}{e^2} \leq \frac{e^{\cos(\omega)}}{Z}$, hence:

$$\dot{\omega} \leq -\frac{2}{e^2} \sin(\omega),$$

which implies, by an ODE comparison theorem:

$$\omega(t) \leq 2 \cot^{-1}(e^{c + 2t/e^2}) \tag{13}$$

where the right-hand side is the solution of $\dot{\gamma} = -\frac{2}{e^2} \sin(\gamma)$ and $c$ depends on the initial condition.

Now consider the initial conditions $\theta_1^\epsilon := \epsilon$ and $\theta_2^\epsilon := \pi - \epsilon$. The corresponding empirical measure is:

$$\mu_\epsilon(0) := \frac{1}{2}(\delta_\epsilon + \delta_{\pi - \epsilon})$$

We want a bound of this kind:

$$\frac{W^1(\mu_\epsilon(t), \mu_0(t))}{W^1(\mu_\epsilon(0), \mu_0(0))} \leq f(t) \quad \forall \epsilon > 0 \quad \forall t \geq 0.$$

for some function $f$.

Notice that $\mu_0(0)$ is a constant solution of our system. The Wasserstein distance w.r.t. the geodesic distance on $\mathbb{S}^1$ is given by:

$$W^1(\mu_\epsilon(t), \mu_0(t)) = \frac{1}{2}(\theta_1(t) + \pi - \theta_2(t)) = \frac{\pi}{2} - \frac{\omega(t)}{2},$$

hence the function $f$ must satisfy:

$$\frac{\frac{\pi}{2} - \frac{\omega(t)}{2}}{\epsilon} \leq f(t) \quad \forall \epsilon > 0 \quad \forall t \geq 0,$$

with initial condition $\omega(0) = \pi - 2\epsilon$. Using the bounds in Eq. (13):

$$\frac{\frac{\pi}{2} - \frac{\omega(t)}{2}}{\epsilon} \geq \frac{1}{\epsilon}\left(\frac{\pi}{2} - \cot^{-1}(e^{c + 2t/e^2})\right)$$

with $c$ s.t. $2\cot^{-1}(e^c) = \pi - 2\epsilon$, i.e. $e^c = \tan(\epsilon)$. Hence it must hold:

$$f(t) \geq \frac{1}{\epsilon}\left(\frac{\pi}{2} - \cot^{-1}(\tan(\epsilon) e^{2t/e^2})\right) \quad \forall \epsilon > 0.$$

Passing to the limit as $\epsilon \to 0$, using a first order Taylor expansion in 0 for $\cot^{-1}(x) \approx \frac{\pi}{2} - x$, we get:

$$f(t) \geq \lim_{\epsilon \to 0} \frac{1}{\epsilon}\left(\frac{\pi}{2} - \cot^{-1}(\tan(\epsilon) e^{2t/e^2})\right) =$$

$$= \lim_{\epsilon \to 0} \frac{\tan(\epsilon)}{\epsilon} e^{2t/e^2} = e^{2t/e^2}.$$

This implies that we cannot improve the exponential time dependence for the stability.

# C  APPENDIX: METASTABILITY

## C.1  PRELIMINARY DEFINITIONS

### C.1.1  SOBOLEV SPACES OF NEGATIVE ORDER

Given a domain $\Omega \subset \mathbb{R}^d$, we define for every $s \in \mathbb{N}$ and $p \in [1, +\infty[$ the corresponding Sobolev space of negative order as the dual:

$$W^{-s,p'}(\Omega) := W_0^{s,p}(\Omega)',$$

where $p' = p/(p-1)$ and $W_0^s(\Omega)$ denotes the closure of $C_c^\infty(\Omega)$ in the usual Sobolev space $W^{s,p}(\Omega)$. The corresponding dual norm for $W^{-s,p'}(\Omega)$ is given by:

$$\|u\|_{W^{-s,p'}(\Omega)} := \sup \left\{ \langle u, v \rangle : v \in W_0^{s,p}(\Omega), \|v\|_{W^{s,p}(\Omega)} = 1 \right\}. \tag{14}$$

For any $s \in \mathbb{N}$ and $p \in [1, +\infty[$ the space $W^{-s,p'}(\Omega)$ is a Banach space. Moreover, if $1 < p < +\infty$, $W^{-s,p'}(\Omega)$ is separable and reflexive.

These definitions can be naturally extended to compact Riemannian manifolds, while the non-compact case is more subtle (the reader can find a reference in (Hebey, 1996)).

Most of our discussion is set within the space $W^{-s,2}(\Omega)$, also referred to as $H^{-s}(\Omega)$. These spaces can be conveniently characterized using Fourier series:

$$H^s(\mathbb{T}^d) = \left\{ u \in C^\infty(\mathbb{T}^n) : \|u\|_{H^s(\mathbb{T}^d)} < \infty \right\},$$

where the norm on $H^s(\mathbb{T}^d)$ is defined as:

$$\|u\|_{H^s_{\mathbb{T}^d}}^2 = \sum_{k \in \mathbb{Z}^d} |\hat{u}_k|^2 (1 + |k|^2)^s.$$

Additionally, note that the Dirac delta $\delta$ is an element of $H^s(\mathbb{R}^d)$ for every $s < -d/2$.

### C.1.2  SPHERICAL HARMONICS AND GEGENBAUER POLYNOMIALS

The classical spherical harmonics can be generalized to higher-dimensional spheres $\mathbb{S}^{d-1}$ in the following way. Consider $\mathbb{P}_l$ the space of homogeneous polynomials $p(x) : \mathbb{R}^d \to \mathbb{C}$ of degree $l$ in $d$ variables, and let $\mathbb{A}_l$ denote the subspace $\mathbb{P}_l$ of harmonic polynomials. Then the spherical harmonics of degree $l$ are an orthogonal basis for:

$$\mathbb{H}_l := \left\{ Y_l : \mathbb{S}^{d-1} \to \mathbb{C} \mid \text{exists } p \in \mathbb{A}_l \text{ such that } Y_l(x) = p(x) \text{ for all } x \in \mathbb{S}^{d-1} \right\}.$$

While explicit formulas can be given by induction, we will just need the following:

**Theorem C.1.** *If $\Delta$ is the Laplace-Beltrami operator on $\mathbb{S}^{d-1}$, then for all $Y_l \in \mathbb{H}_l$, one has:*

$$\Delta Y_l = -l(l + d - 2)Y_l.$$

Furthermore, spherical harmonics have a strict relationship with the convolution defined in Eq. (4), through the so called Gegenbauer polynomials $P_k^\alpha$.

These polynomials, also known as ultraspherical, are orthogonal on $[-1, 1]$ with respect to the probability measure with density $\frac{\Gamma(\alpha+1)}{\sqrt{\pi}\Gamma(\alpha+\frac{1}{2})}(1-t^2)^{\alpha-\frac{1}{2}}$. An useful definition is through Rodrigues formula:

$$P_k^\alpha(t) = \frac{(-1)^k \Gamma(\alpha + \frac{1}{2})}{2^k \Gamma(k + \alpha + \frac{1}{2})} (1 - t^2)^{-\alpha + \frac{1}{2}} \frac{d^k}{dt^k} \left[ (1 - t^2)^{k + \alpha - \frac{1}{2}} \right].$$

They generalize Legendre polynomials and Chebyshev polynomials, and satisfy the so called Funk-Hecke formula:

**Theorem C.2.** *Let* $f : [-1, 1] \to \mathbb{R}$ *such that* $t \to (1 - t^2)^{\frac{d-3}{2}} f(t)$ *is integrable. Then the Funk-Hecke formula holds:*

$$\int_{\mathbb{S}^{d-1}} f(\langle x, y \rangle) Y_l(y) d\sigma(y) = \lambda_l Y_l(x),$$

*with the constant* $\lambda_l$ *given by:*

$$\lambda_l = \int_{-1}^{1} P_l^{\frac{d-2}{2}}(t) f(t) \frac{2\pi^{\frac{d-1}{2}}}{\Gamma(\frac{d-1}{2})} (1 - t^2)^{\frac{d-3}{2}} dt,$$

*where* $\Gamma(\cdot)$ *is the gamma function and* $Y_l$ *is any spherical harmonics on* $\mathbb{S}^{d-1}$ *of degree* $l$.

Other properties are discussed in (Han et al., 2012; Seeley, 1966). Although the nomenclature may not be standardized in the literature, we refer to $\lambda_l$ as Gegenbauer coefficent of $f$ of degree $l$.

## C.2 THE EMERGENCE OF SYMMETRY

For clarity, the following discussion is restricted to $\mathbb{S}^1$ (refer to Section 4.4 for adaptations to higher dimensions). Specifically, notice that identifying each pair of points $x, y \in \mathbb{S}^1$ with the corresponding angle $\theta, \gamma \in [0, 2\pi]$ and using the relation $W(\langle x, y \rangle) = W(\cos(\theta - \gamma))$, the definition of convolution provided in Eq. (4) aligns with the classical notion of convolution on $\mathbb{S}^1$. Furthermore, thanks to this change of variable, the Gengenbauer coefficients $\hat{W}_k$ correspond to the Fourier cosine expansion of $W \circ \cos$. Thus, for simplicity of notation we will continue using the notation $W$ to represent the composition $W \circ \cos$.

**Remark C.3.** *In the case of Transformers, both Assumption 1 and Assumption 2 are satisfied since* $W(\theta) := \frac{1}{\beta} e^{\beta \cos(\theta)} = \sum_{k=0}^{\infty} \frac{2}{\beta} I_k(\beta) \cos(k\theta)$, *where* $I_k(\beta)$ *are the modified Bessel functions of the first kind.*

Consider $\mu_\infty$ as the uniform measure on $\mathbb{S}^1$, and let the intial perturbation be $\mu_0 := \mu_\infty + \rho_0$ in $\mathcal{P}(\mathbb{S}^1)$. If $\mu_t$ evolves according to the following PDE (consequence of Eq. (3)):

$$\begin{cases} \partial_t \mu_t = -\partial_\theta(\mu_t \nabla W * \mu_t), \\ \mu(0) = \mu_0, \end{cases} \tag{15}$$

then the perturbation $\rho_t$ satisfies:

$$\begin{cases} \partial_t \rho_t = \mathcal{L}_{\mu_\infty}(\rho_t) - \partial_\theta(\rho_t \nabla W * \rho_t), \\ \rho(0) = \rho_0, \end{cases} \tag{16}$$

where $\mathcal{L}_{\mu_\infty}(\rho) := -\mu_\infty \Delta W * \rho$ denotes the PDE's linearized operator around $\mu_\infty$.
**Remark C.4.** *Since* $\nabla W * \mu_\infty = 0$, *the uniform measure is an equilibrium of the system.*

The objective of this section is to demonstrate that the dynamics of the system near $\mu_\infty$ are governed by the maximal eigenvalue of $\mathcal{L}_{\mu_\infty}$.

The core estimate is provided by the following lemma:
**Lemma C.5.** *Let* $n \geq 0$. *For any* $g \in H^n(\mathbb{S}^1)$ *and* $f \in W^{n,\infty}(\mathbb{S}^1)$, *the following inequality holds:*

$$\langle f \partial_\theta g, g \rangle_{H^n} \leq C_n \|f\|_{W^{n,\infty}} \|g\|_{H^n}^2.$$

*Proof.* For every $k \leq n$:

$$\langle \partial^k(f\partial g), \partial^k g \rangle_{L^2} = \sum_{j=0}^{k} \binom{k}{j} \langle \partial^{k-j} f \, \partial^{j+1} g, \partial^k g \rangle =$$

$$= \sum_{j=0}^{k-1} \binom{k}{j} \langle \partial^{k-j} f \, \partial^{j+1} g, \partial^k g \rangle + \langle f \partial^{k+1} g, \partial^k g \rangle \leq$$

$$\leq \sum_{j=0}^{k-1} \binom{k}{j} \|\partial^{k-j} f\|_{L^\infty} \|\partial^{j+1} g\|_{L^2} \|\partial^k g\|_{L^2} + \langle f, \frac{1}{2} \partial(\partial^k g)^2 \rangle \leq$$

$$\leq C_k \|f\|_{W^{k,\infty}} \|g\|_{H^k}^2.$$

$\square$

For sufficiently small time intervals, the evolution of $\mu_t$ can be approximated by the solution of the linearized operator. The deviation from this approximation will be estimated in the next paragraph.

### C.3 THE LINEAR PHASE

Dobrushin's estimate applies globally on $\mathbb{S}^1$ and may be suboptimal in the vicinity of the equilibrium $\mu_\infty$. While in general is not possible to have a uniform or polynomial dependence with respect to time (see Appendix B), this estimate can be locally refined by considering the singular values of the linear operator $\mathcal{L}_{\mu_\infty} : H^k(\mathbb{S}^1) \to H^k(\mathbb{S}^1)$. Specifically, under Assumption (1) and Assumption (2), for every $f \in H^k(\mathbb{S}^1)$ with $k \in \mathbb{Z}$ (refer to Lemma E.2):

$$\|\mathcal{L}_{\mu_\infty} f\|_{H^k} \leq \frac{1}{2\pi} \gamma_{max} \|f\|_{H^k}.$$

**Remark C.6.** *Hereafter, $\frac{1}{2\pi}\gamma_k$ will be denoted simply as $\gamma_k$.*

Utilizing the Lagrangian flow, it's possible to bound the growth of the perturbation $\rho_t$ in a small neighborhood of $\mu_\infty$ using $\gamma_{max}$. In particular, the following lemma can be proved:

**Lemma C.7.** *Let $k \geq 0$. There exists $\epsilon > 0$ such that, if $\|\rho_0\|_{H^{-k}} < \epsilon$, then*

$$T_1(\epsilon, \rho_0) := \frac{1}{\gamma_{max}} \ln\left(\frac{\epsilon}{\|\rho_0\|_{H^{-k}}}\right)$$

*is well-defined, and for every $t \in [0, T_1(\epsilon, \rho_0)]$, the following inequality holds:*

$$\|\rho_t\|_{H^{-k}} \leq 3\|\rho_0\|_{H^{-k}} e^{\gamma_{max} t}.$$

**Remark C.8.** *The constant $\epsilon$ is independent of $\rho_0$. Furthermore, this lemma implies that, within the time interval $[0, T_1]$ the evolution of the measure $\mu_t$ remains within $B_{H^{-k}}(\mu_\infty, 3\epsilon)$.*

*Proof.* Given a test function $\bar{\psi} \in H^k(\mathbb{S}^1)$, let $\psi_s$ be the solution of:

$$\begin{cases} \partial_s \psi_s = -\mathcal{L}_{\mu_\infty}(\psi_s) - (\nabla W * \rho_s)\partial_\theta \psi_s & s \in [0, t], \\ \psi_t = \bar{\psi}. \end{cases} \tag{17}$$

Hence, by construction:

$$\begin{aligned} \partial_s \langle \rho_s, \psi_s \rangle &= \langle \partial_s \rho, \psi \rangle + \langle \rho, \partial_s \psi \rangle = \\ &= \langle \mathcal{L}_{\mu_\infty}(\rho_s) - \partial_\theta(\rho_s \nabla W * \rho_s), \psi_s \rangle - \langle \rho_s, \mathcal{L}_{\mu_\infty}(\psi_s) + (\nabla W * \rho_s)\partial_\theta \psi_s \rangle = \\ &= 0. \end{aligned}$$

This implies:

$$\begin{aligned} \|\rho_t\|_{H^{-k}} = \sup_{\|\bar{\psi}\|_{H^k}=1} \langle \rho_t, \bar{\psi} \rangle &= \sup_{\|\psi_t\|_{H^k}=1} \langle \rho_t, \psi_t \rangle = \\ &= \sup_{\|\psi_t\|_{H^k}=1} \langle \rho_0, \psi_0 \rangle \leq \\ &\leq \|\rho_0\|_{H^{-k}} \sup_{\|\psi_t\|_{H^k}=1} \|\psi_0\|_{H^k}. \end{aligned} \tag{18}$$

Next, we need an estimate for $\|\psi_0\|_{H^k}$. The energy of the time-reversed Eq. (17) satisfies:

$$\begin{aligned} \frac{1}{2}\frac{d}{ds}\|\psi_{t-s}\|_{H^k}^2 &= \langle \mathcal{L}_{\mu_\infty}\psi_{t-s}, \psi_{t-s} \rangle_{H^k} + \langle \nabla W * \rho_{t-s}\partial_\theta \psi_{t-s}, \psi_{t-s} \rangle_{H^k} \leq \\ &\leq \gamma_{max}\|\psi_{t-s}\|_{H^k}^2 + C\|\psi_{t-s}\|_{H^k}^2 \|\nabla W * \rho_{t-s}\|_{W^{k,\infty}} \leq \\ &\leq \gamma_{max}\|\psi_{t-s}\|_{H^k}^2 + C\|\psi_{t-s}\|_{H^k}^2 \|\nabla W\|_{H^{2k}}\|\rho_{t-s}\|_{H^{-k}} \leq \\ &\leq (\gamma_{max} + C\|\rho_{t-s}\|_{H^{-k}})\|\psi_{t-s}\|_{H^k}^2, \end{aligned}$$

where the second row is an application of Lemma C.5. Hence, by Gronwall's lemma:

$$\|\psi_0\|_{H^k} \leq \|\psi_t\|_{H^k} e^{\gamma_{max} t + C\int_0^t \|\rho_s\|_{H^{-k}} ds}.$$

Substituting this bound into Eq. (18), we obtain for all $t \geq 0$:

$$\|\rho_t\|_{H^{-k}} \leq \|\rho_0\|_{H^{-k}} e^{\gamma_{max} t + C \int_0^t \|\rho_s\|_{H^{-k}} ds}. \tag{19}$$

The conclusion follows from a standard continuation argument: assume $\|\rho_s\|_{H^{-k}} \leq 3\|\rho_0\|_{H^{-k}} e^{\gamma_{max} s}$. Then, by Eq. (19), in the interval $[0, T_1(\epsilon, \rho_0)]$, it holds:

$$\|\rho_t\|_{H^{-k}} \leq \|\rho_0\|_{H^{-k}} e^{\gamma_{max} t + C \int_0^t \|\rho_s\|_{H^{-k}} ds} \leq$$
$$\leq \|\rho_0\|_{H^{-k}} e^{\gamma_{max} t + 3C\|\rho_0\|_{H^{-k}} \frac{1}{\gamma_{max}} e^{\gamma_{max} t}} \leq$$
$$\leq \|\rho_0\|_{H^{-k}} e^{\gamma_{max} t + 3C\|\rho_0\|_{H^{-k}} \frac{1}{\gamma_{max}} e^{\gamma_{max} T_1(\epsilon, \rho_0)}} \leq$$
$$\leq \|\rho(0)\|_{H^{-k}} e^{\gamma_{max} t + \frac{3C\epsilon}{\gamma_{max}}}.$$

If $\epsilon$ is such that $\frac{3C\epsilon}{\gamma_{max}} \leq 1$, then $\|\rho_t\|_{H^{-k}} < 3\|\rho_0\|_{H^{-k}} e^{\gamma_{max} t}$ in the interval $[0, T_1(\epsilon, \rho_0)]$. This concludes the continuation argument. $\square$

Now we want to control the distance between $\rho$ and the solution $\rho^L$ of the linearized PDE:

$$\begin{cases} \partial_t \rho^L = \mathcal{L}_{\mu_\infty}(\rho^L), \\ \rho_0^L = \rho_0. \end{cases} \tag{20}$$

By considering a loss of regularity we derive the following estimate:

**Lemma C.9.** *Let $k \geq 0$. For all $t \in [0, T_1(\epsilon, \rho_0)]$, the following holds:*

$$\|\rho_t - \rho_t^L\|_{H^{-k-1}} \leq C\|\rho_0\|_{H^{-k}}^2 e^{2\gamma_{max} t}.$$

*Proof.* The difference $\rho - \rho^L$ satisfies the PDE:

$$\partial_t(\rho - \rho^L) = \mathcal{L}_{\mu_\infty}(\rho_t - \rho_t^L) - \partial_\theta(\rho_t \nabla W * \rho_t).$$

Thus,

$$\frac{d}{dt}\|\rho_t - \rho_t^L\|_{H^{-k-1}} \leq \gamma_{max}\|\rho_t - \rho_t^L\|_{H^{-k-1}} + \|\rho_t \nabla W * \rho_t\|_{H^{-k}} \leq$$
$$\leq \gamma_{max}\|\rho_t - \rho_t^L\|_{H^{-k-1}} + \|\rho_t\|_{H^{-k}}\|\nabla W * \rho_t\|_{W^{k,\infty}} \leq$$
$$\leq \gamma_{max}\|\rho_t - \rho_t^L\|_{H^{-k-1}} + C\|\rho_t\|_{H^{-k}}^2.$$

Applying Lemma C.7, we obtain:

$$\frac{d}{dt}\|\rho_t - \rho_t^L\|_{H^{-k-1}} \leq \gamma_{max}\|\rho_t - \rho_L\|_{H^{-k-1}} + C\|\rho_0\|_{H^{-k}}^2 e^{2\gamma_{max} t}.$$

The conclusion is a consequence of Gronwall's lemma (Lemma E.1). $\square$

We can conclude this section with the following proposition, which describe the measure $\mu_t$ at the exit point from $B_{H^{-k}}(\mu_\infty, 3\epsilon)$. In particular, we prove that after a time $T_1$, defined as:

$$T_1(\epsilon, \rho_0) := \frac{1}{\gamma_{max}} \ln\left(\frac{\epsilon}{\|\rho_0\|_{H^{-k}}}\right) \tag{21}$$

the perturbation of the uniform measure consists of a dominant mode characterized by $k_{max}$, along with a remainder made up of two components: the error from discarding the other modes and the error resulting from linearizing the PDE.

**Proposition C.10.** *Let $k \geq 0$. There exists $\epsilon_0 > 0$ such that if $\|\rho_0\|_{H^{-k}} < \epsilon < \epsilon_0$, then:*

$$\mu_{T_1(\epsilon,\rho)} = \mu_\infty + \epsilon \frac{(\hat{\rho}_0)_{k_{max}}}{\|\rho_0\|_{H^{-k}}} \cos(k_{max}\theta) + R(\epsilon, \|\rho_0\|_{H^{-k}}),$$

*where $T_1(\epsilon, \rho_0)$ is defined in Eq. (21). Furthermore, let $\gamma^- := \max_{k \neq \pm k_{max}} \gamma_k$, then:*

$$R(\epsilon, \|\rho_0\|_{H^{-k}}) = O_{H^{-k-1}}\left(\|\rho_0\|_{H^{-k}}\left(\frac{\epsilon}{\|\rho_0\|_{H^{-k}}}\right)^{\frac{\gamma^-}{\gamma_{max}}} + \epsilon^2\right).$$

*Proof.* Notice that the solution to Eq. (20) can be explicitly expressed in terms of the Fourier coefficients of the initial condition:

$$\rho_t^L = \sum_{k \in \mathbb{Z}} (\hat{\rho}_0)_k e^{\gamma_k t} e^{ikx}. \tag{22}$$

Let $\gamma^- := \max_{k \neq k_{max}} \gamma_k$. By substituting Eq. (22) into Lemma C.9 and using the definition of $T_1$, we get:

$$
\begin{aligned}
\rho_{T_1} &= \rho_{T_1}^L + O_{H^{-k-1}}\left(\|\rho_0\|_{H^{-k}}^2 e^{2\gamma_{max}T_1}\right) = \\
&= (\hat{\rho}_0)_{k_{max}} e^{\gamma_{max}T_1} \cos(k_{max}x) + \sum_{k \neq \pm k_{max}} (\hat{\rho}_0)_k e^{\gamma_k T_1} e^{ikx} + O_{H^{-k-1}}\left(\|\rho_0\|_{H^{-k}}^2 e^{2\gamma_{max}T_1}\right) = \\
&= \epsilon \frac{(\hat{\rho}_0)_{k_{max}}}{\|\rho_0\|_{H^{-k}}} \cos(k_{max}x) + \sum_{k \neq \pm k_{max}} (\hat{\rho}_0)_k \left(\frac{\epsilon}{\|\rho_0\|_{H^{-k}}}\right)^{\frac{\gamma^-}{\gamma_{max}}} e^{ikx} + O_{H^{-k-1}}\left(\epsilon^2\right) = \\
&= \epsilon \frac{(\hat{\rho}_0)_{k_{max}}}{\|\rho_0\|_{H^{-k}}} \cos(k_{max}x) + O_{H^{-k-1}}\left(\|\rho_0\|_{H^{-k}} \left(\frac{\epsilon}{\|\rho_0\|_{H^{-k}}}\right)^{\frac{\gamma^-}{\gamma_{max}}} + \epsilon^2\right).
\end{aligned}
$$

$\square$

We turn to our discussion of the subsequent "quasi-linear" phase.

### C.4 THE QUASI-LINEAR PHASE

When the measure $\mu_t$ exits the ball $B_{H^{-k}}(\mu_\infty, 3\epsilon)$, the estimates based on the linearized equation are no longer sufficient to describe the system's evolution. To address this, we build a nonlinear approximation following Grenier's iterative scheme ((Grenier, 2000; Han-Kwan & Nguyen, 2016)).

In the preceding paragraph, we isolated the dominant mode $\cos(k_{max}\theta)$. Let $\alpha > 0$ and consider the solution of the following nonlinear PDE:

$$
\begin{cases}
\partial_t f^\alpha = -\partial_\theta \left(f^\alpha \nabla W * f^\alpha\right) \\
f_0^\alpha = \mu_\infty + \alpha \cos(k_{max}\theta).
\end{cases}
\tag{23}
$$

We will demonstrate the existence of $\delta > 0$, independent of $\alpha$, such that the evolution of $f^\alpha$ within the ball $B_{H^{-k}}(\mu_\infty, \delta)$ is still controlled by $\gamma_{max}$. In particular, we prove in Proposition C.15 that after a time $T_2$:

$$T_2(\delta, \alpha) := \frac{1}{\gamma_{\max}} \ln(\delta/\alpha), \tag{24}$$

the nonlinear evolution of the dominant mode moves outside a neighborhood of the uniform measure, with the radius of this neighborhood $\delta$ being independent of $\alpha$. Moreover, in Proposition C.17, we provide a bound on the distance between $f^\alpha$ and any distribution close to it at time 0, valid over the time horizon $[0, T_2]$.

Let's introduce the following definitions essential for Grenier's scheme:

$$g_1(t, \theta) := e^{\gamma_{max}t} \cos(k_{max}\theta),$$

$$f^{\alpha,K}(t, \theta) := \mu_\infty + \sum_{j=1}^K \alpha^j g_j,$$

where the functions $g_k$ are defined recursively by:

$$
\begin{cases}
(\partial_t - \mathcal{L}_{\mu_\infty})g_k = -\sum_{j=1}^{k-1} \partial_\theta \left(g_j \nabla W * g_{k-j}\right) \\
g_k(0) = 0 \quad \forall k \geq 2.
\end{cases}
\tag{25}
$$

Notice that:

$$\partial_t f^{\alpha,K} + \partial_\theta(f^{\alpha,K} \nabla W * f^{\alpha,K}) = R^{\alpha,K} := - \sum_{\substack{1 \leq j,l \leq K \\ j+l \geq K+1}} \alpha^{j+l} \partial_\theta(g_j \nabla W * g_l). \tag{26}$$

**Lemma C.11.** *Let $K > 0$ and $s \geq 0$. Then $\forall k \leq K$:*

$$\|g_k\|_{H^s} \leq C_K e^{k\gamma_{max}t}.$$

*Proof.* The proof proceeds by induction, showing that $\|g_k\|_{H^{n-k}} \leq C_k e^{k\gamma_{max}t}$, where $n := K + s$.

By definition, we immediately get $\|g_1\|_{H^n} \leq C e^{\gamma_{max}t}$.

For the inductive step, consider $k > 0$ and assume the hypothesis holds up to $k - 1$. Then by Eq. (25):

$$\frac{d}{dt}\|g_k\|_{H^{n-k}} \leq \|\mathcal{L}_{\mu_\infty}g_k\|_{H^{n-k}} + \sum_{j=1}^{k-1} \|\partial_\theta(g_j \nabla W * g_{k-j})\|_{H^{n-k}} \leq$$

$$\leq \gamma_{max}\|g_k\|_{H^{n-k}} + \sum_{j=1}^{k-1} \|g_j \nabla W * g_{k-j}\|_{H^{n-k+1}} \leq$$

$$\leq \gamma_{max}\|g_k\|_{H^{n-k}} + \sum_{j=1}^{k-1} \|g_j\|_{H^{n-k+1}}\|\nabla W * g_{k-j}\|_{W^{n-k+1,\infty}} \leq$$

$$\leq \gamma_{max}\|g_k\|_{H^{n-k}} + \sum_{j=1}^{k-1} \|g_j\|_{H^{n-k+1}}\|\nabla W\|_{L^2}\|g_{k-j}\|_{H^{n-k+1}} \leq$$

$$\leq \gamma_{max}\|g_k\|_{H^{n-k}} + C\sum_{j=1}^{k-1} \|g_j\|_{H^{n-j}}\|g_{k-j}\|_{H^{n-(k-j)}} \leq$$

$$\leq \gamma_{max}\|g_k\|_{H^{n-k}} + C_k e^{k\gamma_{max}t}.$$

Therefore, by Gronwall's lemma (Lemma E.1):

$$\|g_k\|_{H^{n-k}} \leq C_k e^{k\gamma_{max}t}.$$

To conclude, recall the definition of $n$ as $K + s$. $\qquad\square$

**Remark C.12.** *This lemma implies that if $\alpha e^{\gamma_{max}t} \leq \delta < 1$, then:*

$$\|f^{\alpha,K} - \mu_\infty\|_{H^s} \leq C_K \sum_{j=1}^{K}(\alpha e^{\gamma_{max}t})^j \leq \tilde{C}_K(\alpha e^{\gamma_{max}t}),$$

*and similarly:*

$$\|f^{\alpha,K} - \mu_\infty\|_{H^s} \geq \|\alpha g_1\|_{H^s} - C_K \sum_{j=2}^{K}(\alpha e^{\gamma_{max}t})^j \geq \bar{C}_K \alpha e^{\gamma_{max}t}.$$

We now need to estimate the remainder $R^{\alpha,K}$ that appears in Eq. (26):

**Lemma C.13.** *Let $K > 0$, $s \geq 0$ and $\delta < 1$. If $\alpha e^{\gamma_{max}t} \leq \delta$, then:*

$$\|R^{\alpha,K}\|_{H^s} \leq \tilde{C}_K \left(\alpha e^{\gamma_{max}t}\right)^{K+1}.$$

*Proof.* By definition of $R^{\alpha,K}$:

$$\|R^{\alpha,K}\|_{H^s} \leq \sum_{\substack{1 \leq j,l \leq K \\ j+l \geq K+1}} \alpha^{j+l}\|\partial_\theta(g_j \nabla W * g_l)\|_{H^s} \leq$$

$$\leq \sum_{\substack{1 \leq j,l \leq K \\ j+l \geq K+1}} \alpha^{j+l}\|g_j \nabla W * g_l\|_{H^{s+1}} \leq$$

$$\leq C_K \sum_{\substack{1 \leq j,l \leq K \\ j+l \geq K+1}} \alpha^{j+l}\|g_j\|_{H^{s+1}}\|g_l\|_{H^{s+1}}.$$

Finally, Lemma C.11 and the assumption imply:

$$\|R^{\alpha,K}\|_{H^s} \leq C_K \sum_{\substack{1 \leq j, l \leq K \\ j+l \geq K+1}} \alpha^{j+l} e^{(j+l)\gamma_{max}t} \leq \tilde{C}_K \left(\alpha e^{\gamma_{max}t}\right)^{K+1}.$$

□

Now, we can compare the evolution of the dominant mode $f^\alpha$ defined in Eq. (23) and Grenier's approximation $f^{\alpha,K}$. Let $r^{\alpha,K} := f^\alpha - f^{\alpha,K}$.

**Lemma C.14.** *Given $s \geq 0$, there exist constants $K_0$ and $\delta > 0$ (independent of $\alpha$) such that, for every $K \geq K_0$ and for every $t$ such that $\alpha e^{\gamma_{max}t} < \delta$, the following bound holds:*

$$\|r_t^{\alpha,K}\|_{H^s} \leq C(\alpha e^{\gamma_{max}t})^K,$$

*where the constant $C$ depends only on $s$, $K$ and $W$.*

*Proof.* Combining Eq. (23) and Eq. (26), $r^{\alpha,K}$ solves:

$$\partial_t r^{\alpha,K} + \partial_\theta \left(f^{\alpha,K} \nabla W * r^{\alpha,K}\right) + \partial_\theta \left(r^{\alpha,K} \nabla W * f^{\alpha,K}\right) + \partial_\theta \left(r^{\alpha,K} \nabla W * r^{\alpha,K}\right) = R^{\alpha,K}.$$

To study its energy:

$$\begin{aligned}
\frac{1}{2}\frac{d}{dt}\|r^{\alpha,K}\|_{H^s}^2 = &- \langle \partial_\theta \left(f^{\alpha,K} \nabla W * r^{\alpha,K}\right), r^{\alpha,K}\rangle_{H^s} + \\
&- \langle \partial_\theta \left(r^{\alpha,K} \nabla W * f^{\alpha,K}\right), r^{\alpha,K}\rangle_{H^s} + \\
&- \langle \partial_\theta \left(r^{\alpha,K} \nabla W * r^{\alpha,K}\right), r^{\alpha,K}\rangle_{H^s} + \\
&+ \langle R^{\alpha,K}, r^{\alpha,K}\rangle_{H^s}.
\end{aligned} \tag{27}$$

The three terms can be estimated separately:

- first term:
$$\begin{aligned}
-\langle \partial_\theta \left(f^{\alpha,K} \nabla W * r^{\alpha,K}\right), r^{\alpha,K}\rangle_{H^s} = &- \langle \left(\partial_\theta f^{\alpha,K}\right) \nabla W * r^{\alpha,K}, r^{\alpha,K}\rangle_{H^s} + \\
&- \langle f^{\alpha,K} \Delta W * r^{\alpha,K}, r^{\alpha,K}\rangle_{H^s} \leq \\
&\leq C\|f^{\alpha,K}\|_{H^{s+1}}\|r^{\alpha,K}\|_{H^s}^2.
\end{aligned}$$

- second term (using Lemma C.5):
$$-\langle \partial_\theta \left(r^{\alpha,K} \nabla W * f^{\alpha,K}\right), r^{\alpha,K}\rangle_{H^s} \leq C\|f^{\alpha,K}\|_{H^s}\|r^{\alpha,K}\|_{H^s}^2.$$

- third term (again by Lemma C.5):
$$-\langle \partial_\theta \left(r^{\alpha,K} \nabla W * r^{\alpha,K}\right), r^{\alpha,K}\rangle_{H^s} \leq C\|r^{\alpha,K}\|_{H^s}^3.$$

Combining these estimates, and utilizing Lemma C.12 and Lemma C.13:

$$\begin{aligned}
\frac{d}{dt}\|r^{\alpha,K}\|_{H^s} \leq &C(\|f^{\alpha,K}\|_{H^{s+1}} + \|r^{\alpha,K}\|_{H^s})\|r^{\alpha,K}\|_{H^s} + \|R^{\alpha,K}\|_{H^s} \\
\leq &C(\|\mu_\infty\|_{H^s} + C_K\alpha e^{\gamma_{max}t} + \|r^{\alpha,K}\|_{H^s})\|r^{\alpha,K}\|_{H^s} + \\
&+ C_K(\alpha e^{\gamma_{max}t})^{K+1}.
\end{aligned}$$

The conclusion follows via the standard continuation argument.

Assume $\|r^{\alpha,K}\|_{H^s} \leq C\left(\alpha e^{\gamma_{max}t}\right)^K$ and $t \in \left[0, \frac{1}{\gamma_{max}} \ln\left(\frac{\delta}{\alpha}\right)\right]$. If $\delta$ is sufficiently small such that $C_K\delta < 1$, and $K$ is sufficiently large such that $3C < K\gamma_{max}$, then:

$$\begin{aligned}
\frac{d}{dt}\|r^{\alpha,K}\|_{H^s} \leq &C(\|\mu_\infty\|_{H^s} + C_K\alpha e^{\gamma_{max}t} + C_K\left(\alpha e^{\gamma_{max}t}\right)^K)\|r^{\alpha,K}\|_{H^s} + \\
&+ C_K(\alpha e^{\gamma_{max}t})^{K+1} \leq \\
\leq &3C\|r^{\alpha,K}\|_{H^s} + C_K(\alpha e^{\gamma_{max}t})^{K+1}.
\end{aligned}$$

By applying Lemma E.1, which requires the assumption $3C < K\gamma_{max}$, we obtain:

$$\|r^{\alpha,K}\|_{H^s} \leq \frac{C_K \alpha^{K+1}}{\gamma_{max}} e^{(K+1)\gamma_{max}t} \leq \delta C_K e^{K\gamma_{max}t}.$$

Since $\delta < 1$, the solution remains within the desired bound and can be continued. $\qquad\square$

The results presented in this section lead to the following proposition, which provides a lower bound on the growth rate of the dominant mode.

**Proposition C.15.** *Let $f^\alpha$ be the solution of Eq. (23). Given $s \geq 0$, there exists $\delta > 0$ (independent of $\alpha$) such that for $T_2(\delta, \alpha)$ defined as in Eq. (24), the following holds:*

$$\|f^\alpha_{T^2_\alpha} - \mu_\infty\|_{H^s} \geq C\delta. \tag{28}$$

*Proof.* By Lemma C.12 and Lemma C.14, we have:

$$\|f^\alpha - \mu_\infty\|_{H^s} \geq \|f^{\alpha,K} - \mu_\infty\|_{H^s} - \|f^\alpha - f^{\alpha,K}\|_{H^s} \geq$$
$$\geq C_K \alpha e^{\gamma_{max}t} - C_K(\alpha e^{\gamma_{max}t})^K \geq$$
$$\geq C_K \alpha e^{\gamma_{max}t}.$$

To conclude, observe that the maximum $t$ for which the assumption in Lemma C.14 holds is precisely $T_2(\delta, \alpha)$. $\qquad\square$

**Remark C.16.** *Note that the same estimate holds for $s < 0$. Indeed, the only lower bound is needed for $g_1$, which grows as $e^{\gamma_{max}t}$ in every $H^s$ space.*

To conclude this section, we need to establish an estimate on the distance between a given solution $\mu_t$ and the evolution of the dominant mode $f^\alpha_t$.

**Proposition C.17.** *Consider a solution $\mu$ of Eq. (15) with initial condition $\mu_0$, and define $h_t := \mu_t - f^\alpha_t$. If $\|h_0\|_{H^{-k}} = \|\mu_0 - f^\alpha_0\|_{H^{-k}} \leq C\alpha$, then there exists $\delta > 0$ (independent of $\alpha$), such that:*

$$\|h_t\|_{H^{-k}} \leq 3\|h_0\|_{H^{-k}} e^{\gamma_{max}t},$$

*for every $t \in [0, T_2(\delta, \alpha)]$.*

**Remark C.18.** *We will apply this proposition to $\mu_0 = \mu_\infty + \rho(T_1)$.*

*Proof.* Using Eq. (15) and Eq. (23), the distribution $h$ satisfies:

$$\partial_t h = -\partial_\theta(\mu\nabla W * \mu) + \partial_\theta(f^\alpha \nabla W * f^\alpha)$$
$$= -\partial_\theta(h\nabla W * f^\alpha) - \partial_\theta(f^\alpha \nabla W * h) - \partial_\theta(h\nabla W * h)$$
$$= -\mu_\infty \Delta W * h$$
$$\quad -\partial_\theta(h\nabla W * (f^\alpha - \mu_\infty))$$
$$\quad -\partial_\theta((f^\alpha - \mu_\infty)\nabla W * h)$$
$$\quad -\partial_\theta(h\nabla W * h).$$

We can adapt the Lagrangian flow argument used in Lemma C.7 to bound $\|h\|_{H^{-k}}$. Let $\bar\psi \in H^k(\mathbb{S}^1)$, and define $\psi_t$ as the solution of:

$$\begin{cases} \partial_s \psi_s = & \mu_\infty \Delta W * \psi_s \\ & -[\nabla W * (f^\alpha_s - \mu_\infty)]\partial_\theta \psi_s \\ & -\nabla W * [(f^\alpha_s - \mu_\infty)\partial_\theta \psi_s] \\ & -[\nabla W * h_s]\partial_\theta \psi_s \\ \psi_t = \bar\psi. \end{cases}$$

To estimate the energy of $\psi_t$, we proceed as follows:

$$\frac{1}{2}\partial_s\|\psi_{t-s}\|_{H^k}^2 = -\mu_\infty\langle\Delta W * \psi_{t-s}, \psi_{t-s}\rangle_{H^k} +$$
$$+ \langle[\nabla W * (f^\alpha - \mu_\infty)]\partial_\theta\psi_{t-s}, \psi_{t-s}\rangle_{H^k} +$$
$$+ \langle\nabla W * [(f^\alpha - \mu_\infty)\partial_\theta\psi_{t-s}], \psi_{t-s}\rangle_{H^k} +$$
$$+ \langle[\nabla W * h]\partial_\theta\psi_{t-s}, \psi_{t-s}\rangle_{H^k}$$

We estimate each term separately:

- first term (Lemma E.2):

$$-\mu_\infty\langle\Delta W * \psi_{t-s}, \psi_{t-s}\rangle_{H^k} \le \gamma_{max}\|\psi_{t-s}\|_{H^k}^2.$$

- second term (Lemma C.5):

$$\langle[\nabla W * (f^\alpha - \mu_\infty)]\partial_\theta\psi_{t-s}, \psi_{t-s}\rangle_{H^k} \le C\|\psi_{t-s}\|_{H^k}^2\|\nabla W * (f_{t-s}^\alpha - \mu_\infty)\|_{W^{k,\infty}} \le$$
$$\le C\|\psi_{t-s}\|_{H^k}^2\|f_{t-s}^\alpha - \mu_\infty\|_{L^2}.$$

- third term:

$$\langle\nabla W * [(f^\alpha - \mu_\infty)\partial_\theta\psi_{t-s}], \psi_{t-s}\rangle_{H^k} = \langle(f^\alpha - \mu_\infty)\partial_\theta\psi_{t-s}, \nabla W * \psi_{t-s}\rangle_{L^2} +$$
$$+ \langle(f^\alpha - \mu_\infty)\partial_\theta\psi_{t-s}, \nabla^3 W * \psi_{t-s}\rangle_{L^2} +$$
$$+ \langle(f^\alpha - \mu_\infty)\partial_\theta\psi_{t-s}, \nabla^5 W * \psi_{t-s}\rangle_{L^2} + ...$$
$$\le \|f^\alpha - \mu_\infty\|_{L^2}\|\psi_{t-s}\|_{H^1}\|\nabla W * \psi_{t-s}\|_{L^\infty} + ...$$
$$\le C\|f^\alpha - \mu_\infty\|_{L^2}\|\psi_{t-s}\|_{H^1}^2.$$

- fourth term (Lemma C.5):

$$\langle[\nabla W * h]\partial_\theta\psi_{t-s}, \psi_{t-s}\rangle_{H^k} \le C\|\psi_{t-s}\|_{H^k}^2\|h\|_{H^{-k}}.$$

Summarizing, we obtain the following differential inequality:

$$\partial_s\|\psi_{t-s}\|_{H^k} \le (\gamma_{max} + C\|f_{t-s}^\alpha - \mu_\infty\|_{L^2} + \|h_{t-s}\|_{H^{-k}})\|\psi_{t-s}\|_{H^k},$$

and by Gronwall's lemma:

$$\|\psi_0\|_{H^k} \le \|\psi_t\|_{H^k}e^{\gamma_{max}t + \int_0^t C\|f_s^\alpha - \mu_\infty\|_{L^2} + \|h_s\|_{H^{-k}}ds}. \tag{29}$$

Since by construction $\partial_t\langle h_t, \psi_t\rangle = 0$, that implies together with Eq. (29):

$$\|h_t\|_{H^{-k}} := \sup_{\|\bar\psi\|_{H^k}=1}\langle h_t, \bar\psi\rangle = \sup_{\|\psi_t\|_{H^k}=1}\langle h_t, \psi_t\rangle =$$
$$= \sup_{\|\psi_t\|_{H^k}=1}\langle h_0, \psi_0\rangle \le \|h_0\|_{H^{-k}}\sup_{\|\psi_t\|_{H^k}=1}\|\psi_0\|_{H^k}$$
$$\le \|h_0\|_{H^{-k}}e^{\gamma_{max}t + \int_0^t C\|f_s^\alpha - \mu_\infty\|_{L^2} + \|h_s\|_{H^{-k}}ds}.$$

The conclusion follows by a continuation argument. Suppose $\|h_t\|_{H^{-k}} \le 3\|h_0\|_{H^{-k}}e^{\gamma_{max}t}$ and $t \in [0, T_2]$. Then:

$$\|h_t\|_{H^{-k}} \le \|h(0)\|_{H^{-k}}e^{\gamma_{max}t + \int_0^t C\alpha e^{\gamma_{max}s} + 3\|h_0\|_{H^{-k}}e^{\gamma_{max}s}ds} \le$$
$$\le \|h(0)\|_{H^{-k}}e^{\gamma_{max}t + \frac{C\alpha}{\gamma_{max}}e^{\gamma_{max}t}}$$
$$\le \|h(0)\|_{H^{-k}}e^{\gamma_{max}t + \frac{C\delta}{\gamma_{max}}}$$

where the hypothesis $\|h_0\|_{H^{-k}} \le C\alpha$ was used in the second line. By choosing $\delta$ small enough, such that $\frac{C\delta}{\gamma_{max}} \le 1$, we get the desired estimate. $\qquad\square$

### C.5 COMBINING THE TWO PHASES

In Section C.3, we analyzed the initial evolution of a perturbation $\mu = \mu_\infty + \rho$ of the uniform measure by comparing it with the solution of the linearized PDE. After a time $T_1$, the dominant mode emerges. As the perturbation exits the neighborhood $B_{H^{-k}}(\mu_\infty, 3\epsilon)$, a higher order approximation using Grenier's scheme is required to track the nonlinear evolution of this dominant mode, as described in Section C.4. We now consolidate these findings into the following proposition, which we specify to the case of empirical measures.

Let $\{\mu_0^N\}_{N \in \mathbb{N}}$ denote a sequence of random measures in $\mathcal{P}(\mathbb{S}^1)$, where $\mu_0^N$ is the empirical measure associated with $N$ i.i.d. particles sampled from the uniform measure on $\mathbb{S}^1$. Define the radius of the ball in which the linear phase occurs as follows:

$$\alpha^N := \frac{(\widehat{\mu_0^N - \mu_\infty})_{k_{max}}}{N^{1/4}\|\mu_0^N - \mu_\infty\|_{H^{-1}}},$$

and consider the corresponding solutions $\mu_t^N$ and $f_t^N$ of Eq. (15) with initial conditions $\mu_0^N$ and $\mu_\infty + \alpha^N \cos(k_{max}\theta)$, respectively.

**Remark C.19.** *Since $(\widehat{\mu_0^N - \mu_\infty})_{k_{max}} = O(\|\mu_0^N - \mu_\infty\|_{H^{-1}})$, $\alpha_N \to 0$ as $N \to \infty$.*

**Proposition C.20.** *There exist a constant $\delta > 0$ and two sequences of times $T_1^N$ and $T_2^N$ (depending on $\mu_0^N$) such that:*

$$\|f_{T_2^N}^N - \mu_\infty\|_{H^{-2}(\mathbb{S}^1)} > \delta$$

*uniformly in $N$, and*

$$\|\mu_{T_1^N + T_2^N}^N - f_{T_2^N}^N\|_{H^{-2}(\mathbb{S}^1)} \to 0,$$

*as $N \to \infty$ in probability.*

**Remark C.21.** *By Proposition C.15 and Remark C.12 we have both a lower and an upper bound for $\|f_{T_2^N}^N - \mu_\infty\|_{H^{-2}(\mathbb{S}^1)}$, uniform in $N$.*

*Proof.* For clarity, denote $\rho_t^N := \mu_t^N - \mu_\infty$. Applying Proposition C.10 with $\epsilon = N^{-1/4}$ and noting that $N^{1/4}\|\rho_0^N\|_{H^{-1}} \to 0$ in probability (Lemma E.3), we obtain:

$$\mu_{T_1^N}^N = \mu_\infty + \alpha^N \cos(k_{max}\theta) + R^N,$$

where $T_1^N := \frac{1}{\gamma_{max}} \ln\left(\frac{1}{N^{1/4}\|\rho_0^N\|_{H^{-1}}}\right)$ and:

$$R^N = O_{H^{-2}}\left(\|\rho_0\|_{H^{-1}}\left(\frac{\epsilon}{\|\rho_0\|_{H^{-1}}}\right)^{\frac{\gamma^-}{\gamma_{max}}} + \epsilon^2\right).$$

Note that, by Lemma E.3, we have:

$$\frac{1}{\alpha^N}\|R^N\|_{H^{-2}} \to 0 \tag{30}$$

in probability. This implies that the hypothesis of Proposition C.17 are asymptotically satisfied for the initial condition $\mu_{T_1^N}^N$. Therefore, there exists a constant $\delta > 0$ such that $f_t^N$, defined as above, satisfies:

$$\|\mu_{T_1^N + t}^N - f_t^N\|_{H^{-2}} \le 3\|R^N\|_{H^{-2}}e^{\gamma_{max}t},$$

for all $t \in [0, T_2^N]$, with $T_2^N := \frac{1}{\gamma_{max}} \ln\left(\frac{\delta}{\alpha^N}\right)$.

Thus,

$$\|\mu_{T_1^N + T_2^N}^N - f_{T_2^N}^N\|_{H^{-2}} \le 3\frac{\delta}{\alpha^N}\|R^N\|_{H^{-2}} \to 0$$

in probability, as recalled in Eq. (30).

Finally, observe that $\|f_{T_2^N}^N - \mu_\infty\|_{H^{-2}(\mathbb{S}^1)} > \delta$ is an immediate consequence of Proposition C.15. $\square$

**Remark C.22.** *The convergence also holds in Wasserstein distance, as proved in Lemma E.4.*

## C.6 THE CLUSTERING PHASE

In the previous steps, only an approximate understanding of the trajectory of the dominant mode, $f^\alpha$, was necessary. Now, however, we focus on a more specific case. Numerical experiments (see Figure 5) conducted on the Transformers model, $W(\theta) = \frac{1}{\beta} e^{\beta \cos(\theta)}$, reveal that after an initial phase where $f^\alpha$ surpasses a fixed threshold, i.e., $W_1(f_t^\alpha, \mu_\infty) > \delta$, the system converges to $k_{max}$ equally spaced clusters. The time required for the transition from such threshold to the clustered state is independent of $\alpha$. This assumption is summarized in Assumption 3 .

Although this remains an assumption, we believe it holds true, particularly in the case of Transformers. This belief is strongly supported by the fact that probability measures invariant under $k_{max}$-fold rotations form an invariant manifold for the system's dynamics, and the limiting measure must be a sum of Dirac deltas (see Lemma E.5).

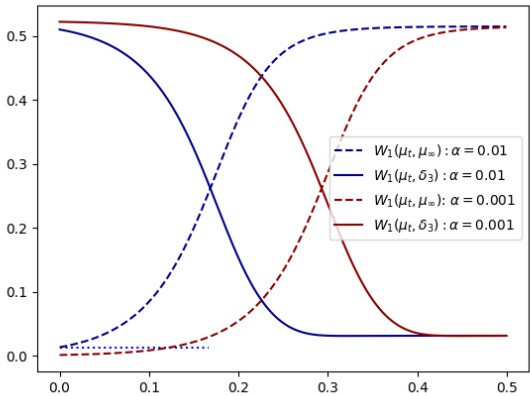

Figure 5: Plots showing the evolution of the Wasserstein distance between $\mu_t$ and the uniform measure $\mu_\infty$ (solid line) and between $\mu_t$ and $\delta_3 := \sum_{k=0}^2 \delta_{2\pi k/3}$ (dashed line). Note that the simulations with the initial condition $f^\alpha$ for $\alpha = 0.01$ and $\alpha = 0.001$ almost completely overlap, differing only by a time shift. This observation supports the validity of Assumption 3.

Under Assumption 3 we can more precisely characterize the final result of the meta-stable phase:

**Theorem C.23.** *Let* $\{\mu_0^N\}_{N \in \mathbb{N}}$ *denote a sequence of random measures in* $\mathcal{P}(\mathbb{S}^1)$, *where* $\mu_0^N$ *is the empirical measure associated with* $N$ *i.i.d. particles sampled from the uniform measure on* $\mathbb{S}^1$. *Under assumptions 1,2 and 3 we can conclude:*

$$\lim_{N \to \infty} \inf_{t \geq 0} W_1(\mu_t^N, \delta^{k_{max}}) = 0,$$

*in probability, where* $\delta^{k_{max}} := \frac{1}{k_{max}} \sum_i \delta_{\frac{2\pi i}{k_{max}}}$ *modulo rotations.*

*Proof.* By Assumption 3, for every $\epsilon > 0$ there exists $\bar{t} > 0$, independent of $N$, such that:

$$W_1(f_{T_2+\bar{t}}^{\alpha^N}, \delta^{k_{max}}) \leq \frac{\epsilon}{2},$$

where $\alpha^N$ and $T_2$ are defined as in Proposition C.20. Therefore, for all $\epsilon > 0$:

$$\lim_{N \to \infty} \inf_{t \geq 0} W_1(\mu_t^N, \delta^{k_{max}}) \leq \lim_{N \to \infty} W_1(\mu_{T_1+T_2+\bar{t}}, \delta^{k_{max}}) \leq$$

$$\leq \lim_{N \to \infty} W_1(\mu_{T_1+T_2+\bar{t}}, f_{T_2+\bar{t}}^{\alpha^N}) + W_1(f_{T_2+\bar{t}}^{\alpha^N}, \delta^{k_{max}}) \leq$$

$$\leq \lim_{N \to \infty} e^{L\bar{t}} W_1(\mu_{T_1+T_2}, f_{T_2}^{\alpha^N}) + \frac{\epsilon}{2},$$

where the classical Dobrushin's estimate (Theorem A.4) is used in the last line, with a finite constant $L$ depending on the properties of $W$. By Proposition C.20 and Remark C.22, it follows that $e^{L\bar{t}} W_1(\mu_{T_1+T_2}, f_{T_2}^{\alpha^N}) \leq \frac{\epsilon}{2}$ definitely in $N$. Hence, the conclusion follows by arbitrariness of $\epsilon$. $\quad\square$

## D  EXTENSION TO HIGHER DIMENSIONS

In this section we discuss what happens if the system is defined on $\mathbb{S}^{d-1}$ by:

$$\begin{cases} \partial_t \mu & = -\operatorname{div}(\mu \nabla(W * \mu)), \\ \mu(0) & = \mu_0, \end{cases} \tag{31}$$

where $W : \mathbb{S}^{d-1} \to \mathbb{R}$ and $W * \mu$ is the convolution defined in Eq. (4).

**Remark D.1.** *The differential operators should be interpreted with respect to the underlying Riemannian manifold structure.*

The uniform measure $\mu_\infty$ remains an unstable equilibrium of the dynamics. Furthermore, the linearized PDE for an initial perturbation $\mu_\infty + \rho_0$ is again:

$$\begin{cases} \partial_t \rho^L & = -\mu_\infty \Delta(W * \rho^L), \\ \rho_0^L & = \rho_0, \end{cases}$$

with $\Delta$ the Laplace-Beltrami operator on the sphere $\mathbb{S}^{d-1}$.

As in the $d = 2$ case, we can study the linear operator $\mathcal{L}_{\mu_\infty}(f) := -\mu_\infty \Delta(W * f)$. Letting $f_{n,j}$ represent the spherical harmonics coefficients, we obtain:

$$(\mathcal{L}_{\mu_\infty}(f))_{n,j} = \mu_\infty n(n + d - 2)\hat{W}_n f_{n,j}.$$

where $\hat{W}_n$ are the Gegenbauer coefficients. This result follows from the fact that spherical harmonics are eigenfunctions of the Laplace-Beltrami operator, combined with the Funk-Hecke theorem for convolutions on spheres (see Section C.1.2). This provides the usual spectral bounds with a similar $\gamma_{max}$ for the linear operator, and an explicit solution $\rho^L$:

$$\rho_t^L = \sum_{n=0}^{\infty} \sum_{j=0}^{Z_n^d} (\rho_0)_{n,j} e^{\gamma_n t} Y_{n,j},$$

where $Z_n^d$ denotes the number of spherical harmonics $Y_{n,j}$ on $\mathbb{S}^{d-1}$ of degree $n$.

It is important to note that we now have a superposition of functions of the same degree $n_{max}$, all growing at the same rate. In particular, the steps outlined in the previous sections can be repeated, but with initial condition for $f^N$ given by:

$$f_0^N = \mu_\infty + \sum_{j=0}^{Z_{n_{max}}^d} \alpha_j^N Y_{n_{max},j}.$$

Additionally, the calculations have to be performed in the space $H^s$ with $s < -d/2$, rather than in $H^{-1}$, the proofs remain analogous.

In $d = 2$, the emerging symmetry was characterized by invariance under rotations by angles of $\frac{2\pi}{k_{max}}$, which was relatively straightforward to describe. However, as we move to higher dimensions, the nature of the symmetry becomes more complex and challenging to express.

Consider the subspace:

$$\mathcal{H}_k := \{\mu \in H^s : \langle \mu, Y_{l,j} \rangle = 0 \quad \forall l \text{ not divisible by } k\}.$$

This set is invariant for the dynamics of Eq. (31), indeed, since $\mathcal{H}_k$ is a vector subspace of $H^s$, it suffices to show that $\partial_t \mu \in \mathcal{H}_k$ for every $\mu \in \mathcal{H}_k$:

$$\begin{aligned} \langle \partial_t \mu, Y_{l,m} \rangle &= -\langle \operatorname{div}(\mu \nabla(W * \mu)), Y_{l,m} \rangle = \\ &= \langle \mu \nabla(W * \mu), \nabla Y_{l,m} \rangle = \\ &= \langle \left( \sum_{n,j} \mu_{n,j} Y_{n,j} \right) \left( \sum_{n',j'} W_{n'} \mu_{n',j'} \nabla Y_{n',j'} \right), \nabla Y_{l,m} \rangle = \\ &= \sum_{n',n,j',j} W_{n'} \mu_{n',j'} \mu_{n,j} \langle Y_{n,j} \nabla Y_{n',j'}, \nabla Y_{l,m} \rangle = \\ &= \sum_{n,j',j} W_{l-n} \mu_{l-n,j'} \mu_{n,j} \langle Y_{n,j} \nabla Y_{l-n,j'}, \nabla Y_{l,m} \rangle, \end{aligned}$$

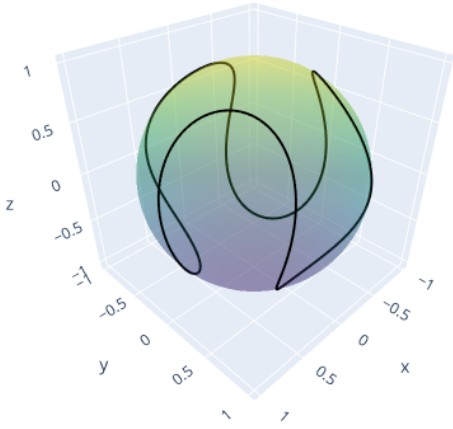

Figure 6: Example of a curve in $\mathcal{H}_3$.

where we used Funk-Hecke theorem and a triple product integral identity for vector spherical harmonics. If this derivative is non-zero, then at least one of the terms in the sum must be non-zero. This implies $l - n = 0 \, mod \, k$ and $n = 0 \, mod \, k$. Thus, $l$ must be a multiple of $k$.

In conclusion:

$$\lim_{N \to \infty} \inf_{t \geq 0} W_1(\mu_t^N, \mathcal{H}_{n_{max}}^\delta \cap \mathcal{P}(\mathbb{S}^{d-1})) \to 0.$$

in probability.

## E    AUXILIARY RESULTS

**Lemma E.1.** *Let $f : [0, \infty) \to \mathbb{R}$ satisfy $f'(t) \leq af(t) + ce^{bt}$ for all $t \in [0, \infty)$, where $0 < a < b$ and $c > 0$. Then:*

$$f(t) \leq f(0)e^{at} + \frac{c}{b-a}e^{bt}.$$

*Proof.* This result follows directly by Gronwall's inequality. Specifically:

$$f(t) \leq f(0) + \frac{c}{b}e^{bt} + \int_0^t \left(f(0) + \frac{c}{b}e^{bs}\right)ae^{a(t-s)}ds =$$

$$= f(0) + \frac{c}{b}e^{bt} + f(0)ae^{at}\int_0^t e^{-as}ds + \frac{c}{b}ae^{at}\int_0^t e^{(b-a)s}ds =$$

$$= f(0) + \frac{c}{b}e^{bt} + f(0)e^{at}(1 - e^{-at}) + \frac{c}{b}\frac{a}{b-a}e^{at}(e^{(b-a)t} - 1) \leq$$

$$\leq f(0)e^{at} + \frac{c}{b-a}e^{bt}.$$

$\square$

**Lemma E.2.** *Let $k \in \mathbb{Z}$. Under Assumptions 1 and 2 the linear operator $\mathcal{L}_{\mu_\infty} : H^k(\mathbb{S}^1) \to H^k(\mathbb{S}^1)$ defined as $\mathcal{L}_{\mu_\infty}(f) := -\mu_\infty \Delta W * f$ satisfies:*

$$\|\mathcal{L}_{\mu_\infty} f\|_{H^k} \leq \frac{1}{2\pi}\gamma_{max}\|f\|_{H^k}.$$

*Proof.* Using the Fourier definition of the spaces $H^k(\mathbb{S}^1)$:

$$\|\mathcal{L}_{\mu_\infty} f\|_{H^k}^2 = \sum_{n \in \mathbb{Z}} n^{2k}|(-\mu_\infty \widehat{\Delta W} * f)_n|^2 =$$

$$= \sum_{n \in \mathbb{Z}} n^{2k}\left(\mu_\infty \gamma_n |(\hat{f})_n|\right)^2 \leq \mu_\infty^2 \gamma_{max}^2 \|f\|_{H^k}^2.$$

$\qquad\square$

**Lemma E.3.** *Let $\{\mu^N\}$ be a sequence of random probabilities on $\mathbb{S}^1$ given by the empirical measures of $N$ i.i.d. samples from the uniform measure. Then $\rho^N := \mu^N - \mu_\infty$ satisfies:*

- $N^{1/4}\|\rho_0^N\|_{H^{-1}} \to 0$,

- $\dfrac{N^{1/4}\|\rho_0^N\|_{H^{-1}}}{(\hat{\rho}_0^N)_{k_{max}}}\left(\|\rho_0^N\|_{H^{-1}}\left(\dfrac{1}{N^{1/4}\|\rho_0^N\|_{H^{-1}}}\right)^{\frac{\gamma^-}{\gamma_{max}}} + (N^{-1/4})^2\right) \to 0$,

*with both of the limits in probability.*

*Proof.* First of all notice that, fixing an orthonormal basis $\{e_k\}_k$ of $L^2(\mathbb{S}^1)$, $\forall \beta > 0$ and $s > \frac{1}{2}$:

$$\mathbb{E}[(N^{\frac{1}{2}-\beta}\|\rho_0^N\|_{H^{-s}})^2] = N^{-2\beta}\mathbb{E}\left[\sum_k Nk^{-2s}\langle\rho_0^N, e_k\rangle^2\right] \leq$$

$$\leq N^{-2\beta}\sum_k Nk^{-2s}\mathbb{E}\left[\left(\frac{1}{N}\sum_{i=1}^N e_k(x_i)\right)^2\right] \leq$$

$$\leq N^{-2\beta}\sum_k k^{-2s} \to 0,$$

where we used Beppo Levi and the indipendence of samples. In particular by Chebychev's lemma this implies $N^{\frac{1}{2}-\beta}\|\rho_0^N\|_{H^{-s}} \to 0$ in probability. Taking $\beta = 1/4$ and $s = 1$ we get the first limit.

Before proving the second limit, observe that

$$\sqrt{N}(\hat{\rho}^N)_k = \sqrt{N}\langle\rho^N, e_k\rangle = \frac{1}{\sqrt{N}}\left(\sum_{j=1}^N e_k(X_j) - \int e_k\right) \to \mathcal{N}(0,1)$$

in distribution by the central limit theorem.

Now:

$$\frac{\|\rho_0\|_{H^{-1}}}{(\hat{\rho}_0)_{k_{max}}}\left(\frac{N^{-1/4}}{\|\rho_0\|_{H^{-1}}}\right)^{\frac{\gamma^-}{\gamma_{max}}-1} + N^{-1/4}\frac{\|\rho_0\|_{H^{-1}}}{(\hat{\rho}_0)_{k_{max}}} =$$

$$= \frac{1}{\sqrt{N}(\hat{\rho}_0)_{k_{max}}}N^{1/2}\|\rho_0\|_{H^{-1}}^{2-\frac{\gamma^-}{\gamma_{max}}}N^{\frac{1}{4}\left(1-\frac{\gamma^-}{\gamma_{max}}\right)} + \frac{N^{1/4}\|\rho_0\|_{H^{-1}}}{\sqrt{N}(\hat{\rho}_0)_{k_{max}}} =$$

$$= \frac{1}{\sqrt{N}(\hat{\rho}_0)_{k_{max}}}\left(N^{\frac{1}{2}+\frac{1}{4}\left(1-\frac{\gamma^-}{\gamma_{max}}\right)}\|\rho_0\|_{H^{-1}}^{2-\frac{\gamma^-}{\gamma_{max}}} + N^{1/4}\|\rho_0\|_{H^{-1}}\right).$$

The terms between the parenthesis converge to zero in probability (indeed $\frac{1}{2} + \frac{1}{4}\left(1 - \frac{\gamma^-}{\gamma_{max}}\right) < \frac{1}{2}\left(2 - \frac{\gamma^-}{\gamma_{max}}\right)$ and we can use the argument above). The term outside the parenthesis instead converges in distribution to $1/Z$ with $Z$ standard normal (because of the continuous mapping theorem, since $1/x$ is almost-surely continuous). Hence, by Slutsky's lemma, we get convergence to $0$ in distribution and then in probability. $\qquad\square$

**Lemma E.4.** *Given two sequences of probability measures $\mu_n, \nu_n$ in $H^{-k}(X)$, with $X$ compact space, such that $\|\mu_n - \nu_n\|_{H^{-k}(X)} \to 0$, then also $W_1(\mu_n, \nu_n) \to 0$ is true.*

*Proof.* Notice that $\|\mu_n - \nu_n\|_{H^{-k}} \to 0$ implies $\langle\mu_n - \nu_n, f\rangle \to 0$ for every $f \in H^k(X)$. Since $H^k(X) \cap C_b(X)$ is dense in $C_b(X)$ (it contains the smooth functions) and since $\mu_n$ and $\nu_n$ are both probability measures, then $\mu_n - \nu_n \rightharpoonup 0$ in weak sense.

Let us fix a subsequence that satisfies $\lim_k W_1(\mu_{n_k}, \nu_{n_k}) = \limsup_n W_1(\mu_n, \nu_n)$. For each $k$, consider $\phi_{n_k} \in Lip_1(X)$ such that $\int \phi_{n_k}(\mu_{n_k} - \nu_{n_k}) = W_1(\mu_{n_k}, \nu_{n_k})$. Without loss of generality,

by adding a constant (which does not affect the integral), we can assume that the functions $\phi_{n_k}$ vanish at the same point, hence they are uniformly bounded and equicontinuous. By Ascoli-Arzelà theorem we can extract a subsequence converging to some $\phi \in Lip_1(X)$. Hence, renaming the subsequence:

$$W_1(\mu_{n_k}, \nu_{n_k}) = \int \phi_{n_k} d(\mu_{n_k} - \nu_{n_k}) \leq 2\|\phi_{n_k} - \phi\|_\infty + \int \phi d(\mu_{n_k} - \nu_{n_k}) \to 0.$$

This concludes the proof. $\qquad\square$

**Lemma E.5.** *Let $W : [-1, 1] \to \mathbb{R}$ be an analytic function and suppose that $\hat{W}_n \neq 0$ for every $n$. Then the equilibria of Eq. (31) can only be the uniform measure or a finite union of submanifolds of dimension at most $d - 2$.*

*Proof.* Consider $\mu_{eq}$ to be an equilibrium of Eq. (31). Then, for every $f \in \mathcal{C}^\infty(\mathbb{S}^{d-1})$, it must hold that:
$$0 = \langle f, \operatorname{div}(\mu_{eq}\nabla(W * \mu_{eq}))\rangle.$$
In particular, substituting $f = \nabla(W * \mu_{eq})$, we obtain:

$$\int_{\mathbb{S}^{d-1}} \|\nabla(W * \mu_{eq})\|^2 d\mu_{eq} = 0.$$

This implies $A := \{x \in \mathbb{S}^{d-1}|\nabla(W * \mu_{eq})(x) = 0\}$ is such that $\mu_{eq}(A^C) = 0$, hence $supp(\mu_{eq}) = A$. Note that $x \to \|\nabla(W * \mu_{eq})(x)\|^2$ is still an analytic function, hence $A$ must be either $\mathbb{S}^{d-1}$ or a finite union of submanifolds of dimension at most $d - 2$ (by Lojasiewicz's structure theorem, (Krantz, 2002)).

To conclude the proof, we just need to study the case $A = \mathbb{S}^{d-1}$. The definition of $A$ implies that $W * \mu_{eq}$ must be constant on $\mathbb{S}^{d-1}$. By the Funk-Hecke theorem, $\forall n > 0, \forall j$:

$$0 = (W * \mu_{eq})_{n,j} = W_n \cdot (\mu_{eq})_{n,j},$$

and thanks to the hypothesis $W_n \neq 0$, we obtain $(\mu_{eq})_{n,j} = 0 \ \forall n \neq 0$ and $\forall j$. Thus, $\mu_{eq}$ is the uniform measure. $\qquad\square$

