# OpenReview forum: "Emergence of meta-stable clustering in mean-field transformer models"
_ICLR.cc/2025/Conference — ICLR 2025 Oral_

### Official Review · Reviewer_DN47 · 2024-10-28

**Soundness:** 4
**Presentation:** 4
**Contribution:** 2
**Rating:** 8
**Confidence:** 2

**Summary:**

The authors consider the evolution of the token distribution in a limit of an infinitely deep transformer which yields a differential equation for the evolution of the tokens. This equation is further simplified by the assumption that the attention parameters are constant over layers, and all equal to the identity, and that there are no MLP subunits. An initial result is that they show that the limit of infinite particles commutes with the time evolution in the sense that there is a PDE that describes the evolution of an empirical distribution with $N$ large enough. From this starting point, they describe a departure from these solutions (initialized so that the limiting measure is the uniform measure)

1) an initial phase where a periodic fluctuation of a certain frequency, $k$, grows
2) a quasi-linear phase where the fluctuations are well-described by the behavior of this mode around the uniform background
3) and finally a collapse phase where the evolution reaches close to a finite set of delta measures.

**Strengths:**

The strengths of this paper are its rigorous analysis of the problem, where they first extract and then rigorously prove that certain clustering dynamics arise directly from attention. This work makes it very clear what structures arise and how they evolve. This is a strong extension of prior work, which relies on less intuitive arguments, and so this paper finds a way to explain a complicated phenomenon more clearly than prior work.

**Weaknesses:**

The main weakness is the lack of connection to a practical setting. The problem considered is highly simplified, and there is little discussion concerning even qualitative arguments explaining what differs in the setting with the greatest interest -- deep transformers with MLP blocks and non-identity weight matrices. It is difficult to understand how to take these results and understand what they say about that setting.

For example:
1. Prior work [1] suggests that the structure of the value, key, and query matrices is important and derives results in various (though simplified) cases
2. The work cited as the foundation for this work [2] discusses the BBGKY hierarchy -- a way to understand the joint law of all the particles by breaking them down in terms of the $n^{\text{th}}$ degree marginals.
3. Other work [3] considers the behavior, including the MLP, and finds (though non-rigorously) that the MLP significantly affects dynamics.

It is of significant interest if connections to ideas like these could be explained in the paper. Of course, the limited scope of one paper may hinder detailed analysis in all possible settings, but it would make it clear how to interpret the results in context.

If the authors could discuss this issue satisfactorily, I would be happy to increase my score for this paper.


**References:**

1) Geshkovski, Borjan, et al. "The emergence of clusters in self-attention dynamics." Advances in Neural Information Processing Systems 36 (2024).
2) Geshkovski, Borjan, et al. "A mathematical perspective on transformers." arXiv preprint arXiv:2312.10794 (2023).
3) Cowsik, Aditya, et al. "Geometric Dynamics of Signal Propagation Predict Trainability of Transformers." arXiv preprint arXiv:2403.02579 (2024).

**Questions:**

1. What is the relationship between this paper and the recent Geshkovski et al. paper "Dynamic metastability in the self-attention model" [1]?
2. Are there simple ways to extend this analysis as $N \to \infty$ in the discrete case? Is there a qualitative difference in this setting?
3. In practice, should we think that transformers operate in phase 1, 2, or 3?
4. How do we know that neglecting the remainder $R$ from equation 5 to 6 is acceptable?
5. What is the interpretation of the $\alpha \to 0$ limit (i.e. should it be interpreted as $N\to\infty$ again)?

**References:**

1. Geshkovski, Borjan, et al. "Dynamic metastability in the self-attention model." arXiv preprint arXiv:2410.06833 (2024).

---

> ### Author Response · Authors · 2024-11-19
>
> We thank the reviewer for their helpful comments and suggestions. We provide answers to the points raised below.
> - **Q1**: The paper "Dynamic Metastability in the Self-Attention Model" is indeed connected to our work in its aim to rigorously explore metastability, as initially suggested in "A Mathematical Perspective on Transformers". However, these two papers approach the problem from complementary perspectives. Geshkovski et al. focus on well-separated configurations as initial conditions, with an interest in the behavior as $\beta\to \infty$ and relatively small token counts $N$ compared to the feature dimension $d$. In contrast, our work initiates from a uniformly distributed configuration, with a particular focus on the limit $N \to \infty$ while keeping $\beta$ finite. This leads us to investigate fundamentally different dynamical regimes and employ distinct mathematical frameworks. Bridging these perspectives would undoubtedly advance the mathematical understanding of transformer architectures, and we are interested in exploring these challenges in future work.
> - **Q2**: The convergence statements in our paper (see Theorem 4.2, 4.3 and 4.5) provide qualitative results highlighting the clustering phenomenon in the $N \to \infty$ limit. It is possible to extend our estimates in order to provide quantitative results showing clustering (with high probability) in the finite $N$ case. However, these estimates are typically technically quite challenging and we opted not to include them in the interest of conciseness and clarity (see also the response to Q3 by reviewer MWwW). Finally, we have understood this reviewer’s question as referring to discreteness in terms of $N$ (number of tokens). If discrete is interpreted as “having a discrete number of layers” (i.e. discrete time in our setup), the validity of our method is justified by Euler’s method for a sufficiently large number of layers and appropriate scaling of the parameters. We hope that we interpreted the reviewer’s comment correctly, but we would be happy to discuss this (and all other points) further during the discussion period.
> - **Q3**: We expect transformers to sequentially operate in all three phases presented in the paper, given enough “time” (depth of the architecture) and large enough $N$. One can, however, classify these phases based on different measures of operational importance. If we examine token movement in terms of spatial displacement, the majority of this occurs during a general version of phase 3, where tokens converge closely to unstable configurations. If we instead consider the duration (in terms of time/layer count) that tokens remain in the neighborhood of certain configurations, such as saddles in the dynamical landscape, then phases 1 and 2 dominate, as they involve longer periods.
> - **Q4**: We agree on the importance of justifying the omission of the remainder term $R$. To address this, starting from Proposition C.17 in the appendix, we provide a rigorous analysis of the growth of $R$ using a Lagrangian approach. This approach enables us to bound the evolution of $R$ in negative Sobolev norms, demonstrating that by the end of the quasi-linear phase, the distance between the original solution $\mu_N$ and the dominant mode $f$ vanishes as $N \to \infty$. In particular, $R$ becomes negligible at every finite time horizon following the quasi-linear phase.
> - **Q5**: As the reviewer suggests, the limit $\alpha \to 0$ is specifically considered because when $N \to +\infty$, the definition provided on lines 321-322 tends toward zero in probability. This is due to the fact that $(\hat{\rho\_0})\_{k\_{\text{max}}} = O(\|\rho_0\|_{H^{-1}})$, which justifies our asymptotic treatment of this parameter in the mean-field context. Attention should be paid, however, to the fact that $f^\alpha$ is only a technical tool in our analysis, representing (a sequence of) leading order approximations of the measure at the exit of the linear phase for increasing values of $N$.

---

> > ### Author Response · Authors · 2024-11-19
> >
> > - **W:** We have briefly addressed the referee’s concern in the main text due to lack of space (lines 522-532). We expand below on what we believe are the main connections of our work with the practical setting, in particular discussing the examples drought up by the referee.
> >    1. Work [1] provides an insightful overview of transformer behavior under various configurations of $Q$, $K$, and $V$. The primary distinction from our setting is the inclusion of layer normalization in our model, which makes it closer to practical implementations but also introduces additional challenges. We believe isolating distinct components of the full model under different sets of simplifying assumptions, as both these articles are doing, is a crucial step toward contributing to the community's broader effort to build a comprehensive understanding of transformer dynamics. Moreover, while our results are presented under the assumption $Q = K = V = \lambda Id$, they extend immediately to the case $Q^T K = \lambda Id$. The gradient flow structure is also preserved for $Q^T K = V$ with a general choice of $V$, suggesting that our analytical tools could be applied to these scenarios.
> > Finally, in [2], numerical experiments on trained transformers observe the emergence of clustering behavior that served as the starting point for our work, pointing to the generic character of this phenomenon.
> >    2. Thank you for highlighting the BBGKY hierarchy; it indeed provides a powerful framework for understanding the joint law of particle systems in physical contexts, which could yield valuable insights on the correlations between a finite number $N$ of tokens and consequently on higher order corrections, for finite $N$, to the mean-field analysis we present. Applying this technical tool, however, requires formulating an Ansatz for the truncation of the hierarchy. Providing an educated guess for such an ansatz, which in turn has important repercussions on the resulting BBGKY analysis, proved to be challenging for the system at hand. To maintain a rigorous mathematical approach, we opted to focus on the mean-field limit, equivalent to the first order truncation of the BBGKY, offering a well-defined approximation without introducing additional implicit assumptions on the $N$-points functions of the system.
> >    3. We agree that including the MLP is important and that incorporating its weights would significantly alter the dynamics. The MLP can indeed be integrated into the model we study by applying the Lie-Trotter splitting scheme, as suggested in the recent paper [3, eq.1.2]. However, since its effect can vary considerably and in the absence of a clear reference choice for $W$, $U$, $b$, selecting the zero matrix allows us to isolate and focus on the self-attention component we aim to study, as it is done typically in the literature in in-context learning [4]. A mathematically rigorous exploration of the interaction between the MLP and self-attention is certainly an important milestone, and we plan to address this in future work. Nonetheless, we consider that focusing on specific components of the model separately is an essential step in advancing the community's broader goal of achieving a more complete understanding of the full transformer dynamics.
> >
> > **References:**
> > 1. Geshkovski, Borjan, et al. "The emergence of clusters in self-attention dynamics." Advances in Neural Information Processing Systems 36 (2024).
> > 2. Geshkovski, Borjan, et al. "A mathematical perspective on transformers." arXiv preprint arXiv:2312.10794 (2023).
> > 3. Geshkovski, Borjan, Philippe Rigollet, and Domènec Ruiz-Balet. "Measure-to-measure interpolation using Transformers." arXiv preprint arXiv:2411.04551 (2024).
> > 4. Sander, Michael E., and Gabriel Peyré. "Towards understanding the universality of transformers for next-token prediction." arXiv preprint arXiv:2410.03011 (2024).

---

> > > ### Comment · Reviewer_DN47 · 2024-11-25
> > > **Reply to Authors**
> > >
> > > Thank you for taking the time to carefully respond to my questions and comments.
> > >
> > > I still believe that the paper does not make a sufficient effort to explain the connection to a more typical setting The extension to $Q^TK=V$ or $Q^TK=\lambda Id$ are not particularly realistic. I have nonetheless increase my score because
> > >
> > > 1. I appreciate that the Authors aim at a mathematically correct paper, and so do not wish to extrapolate,
> > > 2. I feel that the paper is a more clear exposition than other work in this area.,
> > > 3. the mathematical audience for this paper may benefit.
> > >
> > > For these reasons I also lower my confidence in the review.

---

> > > > ### Author Response · Authors · 2024-11-28
> > > >
> > > > We appreciate your constructive comments and the increase in your score. We share your interest in exploring different cases in future works and are grateful for your recognition of the paper’s clarity and mathematical rigor.

---

### Official Review · Reviewer_WDSo · 2024-11-03

**Soundness:** 3
**Presentation:** 3
**Contribution:** 2
**Rating:** 5
**Confidence:** 1

**Summary:**

The authors build on existing work which models the evolution of tokens within a Transformer model. The authors prove the existent of a meta-stable phrase in learning.

Note: I fear I am not the right audience for this paper. I am not at all familiar with the literature, or its relevance in the ML community. I additionally understood little of the content of this paper.

**Strengths:**

- The paper appears to have a high degree of rigor.
- The goal of developing a rigorous theory of how the transformer works is important.

**Weaknesses:**

- It is not clear (to me) what the relevance of the conclusions made by the paper is.

**Questions:**

- Why does the  existence of a "meta-stable phase" imply that then the network learns a richer and more complex representation of the data? What is meant by richer and more complex?

- Are there any clear takeaways from this paper that a practitioner can use to build intuition and how and why a transformer works?

---

> ### Author Response · Authors · 2024-11-19
>
> We appreciate the reviewer’s feedback and we respond to their questions below.
> - **Q1**: The existence of a meta-stable phase implies that the network retains a richer and more complex representation of the data because it avoids the immediate collapse into a single cluster, a phenomenon often associated with a loss of expressivity. This collapse has been identified in trained Transformer models and is viewed as a negative property by practitioners due to its limitation on the network's ability to distinguish between different tokens or features in the data [1,2,3]. In contrast, the metastability phenomenon ensures the persistence of multiple clusters for extended periods, allowing the model to maintain diverse intermediate representations. This diversity supports the encoding of more complex relationships and structures in the data, contributing to the model's overall ability to learn more sophisticated patterns. Our work aims to rigorously describe and prove the appearance of this desirable metastability behavior.
> - **Q2**: The main takeaway for practitioners is that even in our focused analysis of transformer dynamics, the core hyperparameters (such as the number of tokens $N$, the number of layers, the temperature, and the feature dimension $d$) are deeply interdependent. Our findings highlight that adjusting these parameters independently may disrupt the delicate balance required for the model to fully express its dynamics and achieve optimal performance, emphasizing the importance of tuning these parameters in a coordinated way. In particular, the fact that the duration of the metastable phase scales as $O(\ln(N))$ may suggest, in a regime of large context window size in LLMs, a choice of depth of the transformer architecture allowing to maintain ideal expressivity and performance.
>
> **References:**
> 1. Dong, Yihe, Jean-Baptiste Cordonnier, and Andreas Loukas. "Attention is not all you need: Pure attention loses rank doubly exponentially with depth." International Conference on Machine Learning. PMLR, 2021.
> 2. Zhai, Shuangfei, et al. "Stabilizing transformer training by preventing attention entropy collapse." International Conference on Machine Learning. PMLR, 2023.
> 3. Bao, Han, Ryuichiro Hataya, and Ryo Karakida. "Self-attention Networks Localize When QK-eigenspectrum Concentrates." arXiv preprint arXiv:2402.02098 (2024).

---

### Official Review · Reviewer_McN2 · 2024-11-04

**Soundness:** 4
**Presentation:** 4
**Contribution:** 4
**Rating:** 10
**Confidence:** 3

**Summary:**

This paper presents a rigorous investigation of meta-stable clustering phenomena of mean-field Transformer models. By modeling transformer layers as continuous-time flows and employing a mean-field PDE framework, the authors present a very interesting approach to understanding intermediate clustering phases through linearization and higher-order approximation. Numerical experiments support the theoretical findings. This work opens a venue for understanding the meta-stable dynamics of Transformer models.

**Strengths:**

1. I appreciate the authors provide rigorous proofs that illustrate the convergence of the tokens to structured meta-stable manifolds, and identifies the influence of temperature, which is a key factor in understanding the mechanism of transformers. In particular, understanding the mean-field transformer model is beneficial for tasks involving long-context dependencies.

2. Numerical experiments are presented to demonstrate that the theoretical predictions of the clustering dynamics align well with observed practical results. It is nice to see the number of clusters is influenced by the temperature, which I found surprising.

3. Overall, the paper is well-written, and the results are sound. The decomposition of the transformer dynamics into linear, quasi-linear, and collapsing phases allows for a systematic analysis of the clustering phenomena. I'm also excited to see new PDE techniques (the Grenier's scheme) applied to the analysis of transformer dynamics. The findings are both highly relevant to the field and of significant interest to the community.

**Weaknesses:**

1. (Minor) In the contribution bullet point 3, the authors state that "the periodicity developed in the first phase is maintained over exponentially long time intervals." The authors may want to clarify in what sense the time interval is "exponential long" and where in the paper this claim is made.

2. (Minor) Page 10, Line 504, Should "Figure 4" be "Figure 3"?

3. (Minor) Page 5, Line 233, should the coefficient $\hat{W}_k$ be $\tfrac{1}{\beta}I_k(\beta)$?

**Questions:**

I only have a minor question. The authors make a key Assumption 3 regarding convergence to the clustering phase. While they discuss this assumption based on (Markdahl et al., 2017), relying on periodicity, I am curious whether it would still hold in the generic case--specifically, in the case where the initial condition in equation (6) contains multiple modes. Could the authors provide some intuition and discuss potential implications of their result in this case?

---

> ### Author Response · Authors · 2024-11-19
>
> We thank the reviewer for their detailed review and the points raised, which have improved the paper. We address each of them in the following responses.
>
> - **W1+W2+W3**: Thank you for your careful reading and for pointing this out, we apologize for these oversights. To clarify, the dominant mode developed in the first phase continues to grow exponentially in time and its periodicity is maintained over time intervals of length $O(\ln N)$. We updated the wording at lines 69-70 to reflect this more precise characterization, and made the suggested corrections at lines 504 and 233.
> - **Q1**: The delicate aspect of this assumption lies in the fact that a $t \to \infty$ limit is taken: without this assumption, our results show that even in the presence of small perturbations of the uniform measure (activating potentially all Fourier modes), the linear phase naturally isolates a dominant mode, allowing us to recover equation (6) and, under Assumption 3, convergence to $k_{max}$ clusters (in the joint limit $N \to \infty$ and $t_N \to \infty$).
> In the $t \to \infty$ limit, we proceed to discuss possible extensions of our Assumption 3 to the multimodal setting:
>   - In general, for an initial condition with a distribution that combines multiple modes, the dynamics restrict to distributions with periodicity given by the greatest common divisor $k_{gcd}$ of the modes. Proving that all the initial conditions perturbing the uniform measure with $k_{gcd}$ periodicity converge to $k_{gcd}$ equally spaced clusters is, however, harder than proving that Assumption 3 holds, since the set of initial conditions on which the convergence property should be proven is strictly larger than the ones in Assumption 3. Note that it is easy to construct fixed points of the dynamics that are composed of multiple, non-identical clusters. It is unclear to us what the basins of attractions of these stationary distributions are and, as a consequence, it is difficult to develop an intuition on the veracity of the more general version of Assumption 3 considered in this paragraph (more difficult than in the case of Assumption 3 as written in the paper, which was corroborated by numerical simulations).
>   - On the other hand, one could ask themselves the converse question: are there initial conditions perturbing the uniform measure with modes that do *not* have $k^*$ periodicity but still leading to $k^*$ clusters as $t \to \infty$, i.e., “creating” periodicity? While we strongly suspect that this phenomenon cannot occur in our system (if at all, it can only occur in a set of initial conditions of measure 0), a proof of this fact would require a more in-depth understanding of the PDE dynamics.
> The *direct* implications of the above on our analysis would be relatively limited, given that in the linear phase we know that only one mode is generically selected by the dynamics. However, indirectly, a more in-depth understanding of the structure of the basins of attraction of this PDE would have far-reaching implications in the study of its dynamics, for instance in allowing to prove that Assumption 3 actually holds.

---

> > ### Comment · Reviewer_McN2 · 2024-11-25
> > **Thank you for the response!**
> >
> > I have carefully reviewed the author's thoughtful response along with the other reviews. I think this is a strong and well-written paper that offers a new perspective on transformer dynamics.

---

> > > ### Author Response · Authors · 2024-11-28
> > >
> > > Thank you for your kind and positive comments. We deeply appreciate your acknowledgment of our contributions and your insights throughout the review process.

---

### Official Review · Reviewer_bdQd · 2024-11-04

**Soundness:** 3
**Presentation:** 3
**Contribution:** 3
**Rating:** 8
**Confidence:** 2

**Summary:**

Transformers are performant but not theoretically well understood. In particular, it is not clear how to think about the dynamics associated with a large stack of transformer layers. Geshkovski et al. (2023) made a formal analogy between these dynamics and the dynamics of a mean-field interacting particle system, and were able to characterize various aspects of these dynamics, at least in a simplified theoretical setting. The authors of this work build upon this earlier work and characterize not just the long-time behavior, but associated metastable dynamics, which may be important in practice.

**Strengths:**

The authors present a detailed mathematical analysis of a simplified model of transformers, and rigorously prove a variety of results. They are good about presenting their results at a high level in the main text, and leaving the fine technical details in SI. They closely connect their approach to a variety of recent work, principally the aforementioned work by Geshkovski et al.

**Weaknesses:**

The authors present their results clearly and do not appear to overstate them. My major concern is mostly that the authors' results concern kind of a toy setting somewhat far from the kinds of transformers people use in practice. One assumes Q = K = V = Id (line 123), one takes a limit where the number of layers go to infinity, the number of tokens $N$ is assumed to go to infinity, etc. This makes sense given that theoretical analysis of any kind is quite difficult, but makes it unclear to what extent the authors' results shed light on transformers 'in the wild'. It would be extremely helpful if the authors could provide some numerical experiments for slightly more realistic transformer models that show key predictions from their theory hold, at least approximately.

Also, the authors could provide additional commentary on what happens if one tries to generalize their results. What is expected if Q, K, and V are not all the identity? How does one expect finite $N$ corrections to behave, and what do they change about dynamics?

**Questions:**

1. Do slightly more realistic transformers behave in a way consistent with the predictions of this theory? Relatedly, if the theory assumes Q = K = V = Id, what is the right point of comparison for a language model?
2. How does the theory change if certain assumptions (e.g., Q = K = V = Id, finite $N$) are modified or generalized?

---

> ### Author Response · Authors · 2024-11-19
>
> - **Q1 + Q2 + W**:  We sincerely appreciate the reviewer's thoughtful comments regarding the theoretical assumptions in our work. While our model makes several simplifying assumptions, these choices enable rigorous analysis of fundamental transformer dynamics that would be difficult to isolate, at present, in more complex settings.
> We note that  the case $Q = K = V = \lambda Id$ is not the only situation in which our results apply. Indeed, our results extend immediately to the case $Q^T K = \lambda Id$. Furthermore, the gradient flow structure is preserved under the assumption $Q^T K = V$ for a general choice of $V$, suggesting our analytical tools could be extended to these settings. Our interest in developing a theoretical analysis of clustering phenomena is also motivated by empirical observations in the reference work [1], which performed experiments on more realistic transformer architectures [2], and is further supported by a long line of research on token uniformity, oversmoothing, and rank collapse (see, for example, the references in the answer to reviewer WDSo or in [1]). This suggests that, while the mechanisms we analyze theoretically may not capture all aspects of transformer behavior, they isolate key components that are fundamental to the dynamics of transformers. We have briefly summarized this discussion in the conclusions, specifically on lines 522–532, of the new version of the paper for clarity.
> While we are actively investigating extensions including more general weight matrices and MLPs, we believe that, by clearly isolating specific dynamical phenomena in a tractable setting, our current theoretical contribution could provide a foundation for understanding how additional complexities interact with these base dynamics, thereby representing an important stepping stone toward the open problem of analyzing more general architectures.
>
>
> **References:**
> 1. Geshkovski, Borjan, et al. "A mathematical perspective on transformers." arXiv preprint arXiv:2312.10794 (2023).
> 2. Lan, Z. "Albert: A lite bert for self-supervised learning of language representations." arXiv preprint arXiv:1909.11942 (2019).

---

> > ### Comment · Reviewer_bdQd · 2024-12-03
> >
> > I thank the authors for their thoughtful responses to my review and to others' reviews. I still think that the setting considered in the paper is kind of toy, but also acknowledge that doing theory in practice requires many simplifying assumptions. I think the paper is a solid contribution to the literature. I've raised my score.

---

> > > ### Author Response · Authors · 2024-12-03
> > >
> > > Thank you for your response and for increasing your score. We appreciate your consideration of our work.

---

### Official Review · Reviewer_MWwW · 2024-11-09

**Soundness:** 3
**Presentation:** 3
**Contribution:** 3
**Rating:** 8
**Confidence:** 3

**Summary:**

This paper investigates the evolution of tokens in simplified transformer models using a mean-field formulation. By analyzing the long-term behavior of interacting particle systems, the authors prove results on metastable clustering, demonstrating that the model remains close to a metastable manifold of solutions. The study identifies the existence of linear, quasi-linear, and clustering phases in the token evolution.

**Strengths:**

The authors provide a comprehensive analysis of metastability that precisely describes the evolution of tokens. They successfully identify the existence of linear, quasi-linear, and clustering phases, contributing valuable insights into the behavior of simplified transformer models.

**Weaknesses:**

- The model considers all attention parameters $Q, K, V$ as fixed identity matrices. This significant simplification may limit the applicability of the theoretical results, as it does not capture the complexity of learned attention mechanisms.

- Some notations in the paper are not properly defined, which can hinder readers' understanding of the results. For instance, the notation $\hat{\rho}\_0$ related to Theorems 4.2 and 4.3 is not introduced. Additionally, $\mu\_{cluster}$ in Equation (8) seems to be defined on a one-dimensional space since $2\pi j / k_{max}$ is a scalar, making it unclear how to measure the W1 distance from $f_*(t)$. Providing clearer definitions and explanations would enhance the paper's readability.

- For sufficiently large $N$, the parameter $T_1$ becomes negative. This could restrict the applicability of the theorem to certain token counts. Clarification on how this affects the results and whether it imposes practical limitations would be beneficial.

- Minor: At the first glance, I thought the propagation of chaos result (Theorem 3.1) is also one of the main contributions but is actually weak due to the exponential growth of the bound. Now I see that this result serves to emphasize the importance of the metastability analysis. Clarifying this role could prevent potential confusion for readers.

**Questions:**

- In the integral representation of (USA-MF) (lines 148-149), the coefficient $1/N$ seems redundant since $\mu$ is an empirical distribution which contains the normalization term. Also, I’m not sure if (SA-MF) representation is also appropriate because of the same reason.

- Taking sufficiently large $N$, $T_1$ becomes negative. Does this mean the non-existence of linear phase?

- The convergence rate of W1 distance in Theorem 4.3 is not given in the main text. Can you clarify it?

---

> ### Author Response · Authors · 2024-11-19
>
> Thank you for your thoughtful and detailed review. We appreciate your insights and feedback. Below, we address each of your questions and comments.
>
> - **Q1**: You are correct, and we appreciate your careful reading. The coefficient $\frac{1}{N}$ in equation (USA-MF) is indeed redundant, as it is implicitly included in the empirical distribution. This typo has been corrected at lines 148-149. The (SA-MF) representation remains appropriate due to the definition of $Z_{\beta, \mu}$ which ensures normalization.
>
> - **Q2 + W3**: We understand the concern about the well-definedness of $T_1$​ when the parameter becomes negative. This point is addressed by noting that $\rho_0 = \mu_N - \mu_{\infty}$ implicitly depends on $N$ and that, as a result of the Central Limit Theorem, when $N \to \infty$, the $H^{-1}$ norm of $\rho\_0$ scales as $N^{-1/2}$ (Lemma E.3). Consequently, even with large values of $N$, this scaling ensures that the linear phase remains well-defined. We have clarified this point in the revised version at lines 308-309 and explicitly specified that the dependence of $\rho_0$ on $N$ was omitted for simplicity at lines 302-303.
>
> - **Q3**: Our approach does not provide a specific convergence rate for the $W_1$​ distance to keep the proof focused and readable. Instead, we rely on a qualitative argument: weak convergence on compact metric spaces implies $W_1$ convergence. However, a quantitative bound for the convergence rate could indeed be provided: the steps in our proof could yield an estimate in a negative Sobolev norm setting, showing a polynomial rate in $\frac{1}{N}$. We expect that by adapting techniques from [1], one could derive from such results a quantitative upper bound in $W_1$. Nonetheless, we believe that the technical details required by such an adaptation would obscure the main insights of the paper, and we opted to omit such a bound in the interest of readability and conciseness.
>
> - **W1**: While our model simplifies certain aspects, these assumptions allow us to isolate key phenomena in transformer dynamics, such as metastability and clustering, which would be difficult to study rigorously in more complex settings. We believe our analysis provides a foundation for future work, both theoretical and empirical, on more general transformer architectures.  We elaborate further on these considerations in our response to Reviewer bdQd and have added a comment in the conclusions of the revised version of the paper (lines 522–532).
> - **W2 + W4**: We appreciate your feedback on the notation, particularly around Theorems 4.2 and 4.3. We have now incorporated the definition of $(\hat{\rho_0})\_{k\_{max}}$ as the Fourier coefficient of $\rho\_0$ with index $⁡k_{max}$​ (lines 314-315). Since we present every statement in that section for $d=2$, we have also clarified the identification of $S^1$ with the interval $[0, 2\pi]$ in defining $\mu_{\text{cluster}}$​ (lines 296-297). Concerning Theorem 3.1, we have emphasized on lines 62-63 that it justifies and supports the more refined metastability analysis carried out in the rest of the paper. However, we stress that this bound is later combined, during the nonlinear dynamical phase 3, with the more refined estimates we provide to complete the dynamical picture presented in our paper.
>
> **References**:
> 1. Peyre, Rémi. "Comparison between W2 distance and Ḣ− 1 norm, and localization of Wasserstein distance." ESAIM: Control, Optimisation and Calculus of Variations 24.4 (2018): 1489-1501.

---

> ### Comment · Reviewer_MWwW · 2024-11-26
>
> Thank you for the clarification and for revising the manuscript. My concerns have been addressed, and each statement is now clear and understandable. While I still find the model somewhat restrictive, I acknowledge that this work provides an interesting interpretation of the transformer model. Therefore, I have increased the score to 8.

---

> > ### Author Response · Authors · 2024-11-28
> >
> > Thank you for your valuable feedback and for increasing the score. We’re glad our clarifications addressed your concerns and appreciate your recognition of our work.

---

### Author Response · Authors · 2024-11-28

We thank the reviewers once again for their suggestions. We have implemented their recommendations, corrected the typos, and added some additional references in the revised version we just uploaded. We are happy to address any further comments or questions during the remaining days of the discussion period.

---

### Meta-Review · Area_Chair_hq1a · 2024-12-18

**Metareview:**

The authors studied a mean-field limit for a simplified model of transformers, where the number of tokens and the depth of the network approaches infinity. While the framework was first introduced in Geshkovski et al. (2023), the authors contributed several technical results for the model. This includes a propagation of chaos result justifying the mean-field limit, and characterizing the different phases of the token dynamics.

As many reviewers have mentioned, the main weakness of this work is the simplified setting, and it is questionable whether this analysis will eventually yield useful insights towards a practical transformer. However, as most of the reviewers also recognized, this paper has made clear and precise mathematical contributions towards understanding the toy model, which is already quite technically challenging in itself. I would interpret studying the toy model as a necessary step towards understanding a practical transformer architecture, and hence the contributions here are welcomed.

Given the high average rating, and the clear contributions, I would recommend accept.

**Additional Comments On Reviewer Discussion:**

The discussions with reviewer MWwW and bdQd have made progress towards increasing the score. Here the discussions revolved around
1. The simplicity of the toy model.
2. The clarity of the notation.
3. The strength of the propagation of chaos result.

While the first concern remains essentially unresolved, both reviewers agreed that the results remains a solid contribution towards understanding transformers, as the current setting is already non-trivial to analyze.

---

### Decision · Program_Chairs · 2025-01-22

Accept (Oral)